# A quinary WTaCrVHf nanocrystalline refractory high-entropy alloy withholding extreme irradiation environments

O. El Atwani [1] ✉, H. T. Vo[1], M. A. Tunes [1], C. Lee[2,3], A. Alvarado [4,5],
N. Krienke[6], J. D. Poplawsky [7], A. A. Kohnert [1], J. Gigax[2], W.-Y. Chen[8], M. Li[8],
Y. Q. Wang [1], J. S. Wróbel [9], D. Nguyen-Manh [10,11], J. K. S. Baldwin[2],
O. U. Tukac[12], E. Aydogan[12], S. Fensin[2] & E. Martinez[5]

In the quest of new materials that can withstand severe irradiation and mechanical extremes for advanced applications (*e.g.* fission & fusion reactors, space applications, etc.), design, prediction and control of advanced materials beyond current material designs become paramount. Here, through a combined experimental and simulation methodology, we design a nanocrystalline refractory high entropy alloy (RHEA) system. Compositions assessed under extreme environments and in situ electron-microscopy reveal both high thermal stability and radiation resistance. We observe grain refinement under heavy ion irradiation and resistance to dual-beam irradiation and helium implantation in the form of low defect generation and evolution, as well as no detectable grain growth. The experimental and modeling results—showing a good agreement—can be applied to design and rapidly assess other alloys subjected to extreme environmental conditions.

Clean energy production is the cornerstone of our time. Options for sustainable clean energy include advanced power generation systems that have the potential to drastically reduce the emission of greenhouse gases. These advanced systems are often required to operate under harsh conditions to optimize efficiency, which poses several challenges for the available materials. An example of advanced power system is one that is associated with fusion energy[1]. Beyond traditional fission-based systems, fusion reactors not only promise nearly unlimited clean energy, but also avoid the generation of long-life radioactive waste produced in fission devices. One remaining challenge is that of materials, which can withstand extreme conditions of radiation, temperature, and stress, with long-term steady properties for the power

plant to be economically viable[2–4]. A key component in current tokamak designs is the divertor, which will be in contact with the deuterium-tritium (D-T) plasma and sustain severe fluxes of particles (helium (He) ash, D and T) and heat, along with radiation damage induced by high-energy neutrons[2,3]. Tungsten (W) is the current element of choice for the plasma-facing components (PFCs) due to its beneficial properties in terms of heat conduction, mechanical response, and T retention[5,6]. However, He bubble formation, surface morphology evolution, and neutron damage compromise its ability to reach the viability requirements[7–11]. Several strategies have been proposed to enhance the properties of the material facing the plasma. Reducing the grain size, hence increasing the density of interfaces, is

[1]Materials Science and Technology Division, Los Alamos National Laboratory, Los Alamos, NM, USA. [2]Center for Integrated Nanotechnology, Los Alamos National Laboratory, Los Alamos, NM, USA. [3]Department of Materials and Mechanical Engineering, Auburn University, Auburn, AL, USA. [4]Theoretical Division, Los Alamos National Laboratory, Los Alamos, NM, USA. [5]Departments of Mechanical Engineering and Materials Science and Engineering, Clemson University, Clemson, SC, USA. [6]Materials Science and Engineering, University of Wisconsin-Madison, Madison, WI, USA. [7]Materials Science and Technology Division, Oak Ridge National Laboratory, Oak Ridge, TN, USA. [8]Division of Nuclear Engineering, Argonne National Laboratory, Lemon, IL, USA. [9]Faculty of Materials Science and Engineering, Warsaw University of Technology, ul. Wołoska, 02–507 Warsaw, Poland. [10]Culham Center for Fusion Energy, United Kingdom Atomic Energy Authority, Abingdon OX14 3DB, UK. [11]Department of Materials, University of Oxford, Oxford OX1 3PH, UK. [12]Metallurgical and Materials Engineering, Middle East Technical University, Ankara, Turkey. ✉e-mail: osman@lanl.gov

one of them[12]. Grain boundaries are known to promote defect annihilation and therefore, decrease the overall amount of defects generated by irradiation, which leads to deleterious effects on the material properties[13,14]. However, this approach can suffer from some drawbacks in pure materials such as the thermal instability of the nanocrystalline grains (coarsening at the application temperature)[15]. Another approach is to develop alloys where elements can increase strength, act as defect annihilation and recombination sites, and enhance the thermal stability of the material. Recently, a novel set of alloys based on equiatomic compositions of several principal elements (multi-principal elements alloys (MPEAs) or high-entropy alloys (HEA)) have been developed[16,17]. The configurational entropy of mixing in multicomponent alloys tends to be the major thermodynamic driving force to stabilize the solid solution based on simple underlying face-centered cubic (FCC) or body-centered cubic (BCC) crystalline structures[18]. Equiatomic compositions maximize the entropic term of the Gibbs free energy of mixing, promoting the formation of random solutions versus intermetallic phases or phase decomposition[19].

W-based refractory HEAs (RHEAs) have been recently developed in the context of high-temperature applications, showing high melting temperature (above 2873 K) and superior mechanical strength at high temperatures compared to Ni-based superalloys or pure W[20,21]. Combining the two approaches above, the authors have recently developed a refractory low-activation HEA based on W-Ta-Cr-V[22]. Its response to loop formation under ion irradiation[22] and He implantation[23] is enhanced compared to previously developed W-based alloys, showing no noticeable dislocation loop formation and smaller He bubbles with no radiation-induced segregation at grain boundaries upon heavy ion irradiation and He implantation, respectively. However, this quaternary RHEA demonstrated Cr- and V-rich precipitates which could be detrimental to mechanical properties in terms of embrittlement.

In this study, a design strategy to further improve the overall response of the W-Ta-Cr-V RHEA is introduced and resulted in the development of a quinary RHEA with improved resistance to high-temperature and irradiation environments. The aim of such design is to develop a RHEA with higher irradiation resistance, high thermal stability, enhanced control over the morphology, and no precipitation at reactor-relevant temperatures. By using a combination of (i) thermophysical parameter calculations including minimizing the valence electron concentration (VEC) which was shown to enhance the mechanical performance of HEAs[24,25], (ii) the CALculation of PHAse Diagrams (CALPHAD) method with its newest high-entropy alloy database, and (iii) Cluster Expansion (CE) along with Monte Carlo calculations to predict the alloy's morphology, a methodology to foster the discovery of novel RHEAs for application in extreme environments is proposed. Results from developing such a material system (addition of Hf to the W-Ta-Cr-V RHEA) manufactured via magnetron-sputtering deposition with nanocrystalline grains are discussed. Computational thermodynamics approaches such as the CALPHAD method and density-functional theory-informed CE formalisms are used to select optimal compositions and predict thermodynamic properties. These compositions are then manufactured experimentally to test both their ion irradiation and hardness responses, aiming at validating modeling predictions. It is worth emphasizing that the irradiation response, thermal stability, strength, and morphology predictions are studied here. The results show that the quinary RHEA has microstructural stability along with promising ion irradiation response upon single and dual-beam ion irradiation conditions. This material response can be attributed to a combination of factors, including the high density of stable grain boundaries (even showing grain refinement), chemical complexity altering defect recombination rates, and a decrease in the order-disorder transition temperature (ODTT) as compared to the original four elements RHEA. The described design methodology can be further utilized to devise and synthesize new RHEAs for different applications.

## Results

The design stage of the quinary HEA started with the calculation of the thermophysical parameters, mainly enthalpy of mixing ($\Delta H_{mix}$), entropy of mixing ($\Delta S_{mix}$), atomic size difference ($\delta$), and the omega parameter ($\Omega$) for several compositions. The ability of a material to form a single-phase was empirically shown to depend on these three parameters. For that, the value for $\Delta H_{mix}$ was recommended to be in the range of −15 to 5 kJ per mole, while $\delta$ should be smaller than 6.6 and $\Omega$ larger than 1.1[26,27]. $\Delta H_{mix}$, $\delta$, and $\Omega$ were then calculated using Eqs. 1−4 respectively. Other theoretical parameters such as density ($\rho$), melting temperature ($T_m$), and VEC were calculated using the rule of mixture, as shown in Eq. 5, where $i$ represents the element type, $X$ represents the property, $r$ is the atomic radius, and $c$ represents the concentration of element $i$.

$$\Delta H_{mix} = \sum_{i<j} 4 H_{ij}^{mix} c_i c_j \tag{1}$$

$$\delta = 100 \sqrt{\sum_{i=1}^{N} c_i (1 - r_i/r)^2} \tag{2}$$

$$\Omega = T_m \Delta S_{mix} / |\Delta H_{mix}| \tag{3}$$

$$\Delta S_{mix} = -R \sum_{i=1}^{N} c_i \ln c_i \tag{4}$$

$$X = \sum c_i (X)_i \tag{5}$$

The atomic radii and enthalpy used in the calculations are taken from refs. 28,29, respectively. The base material for this design was the W-Ta-Cr-V HEA which shows high mechanical strength and radiation resistance to loop and He bubble formation under ion irradiation[22]. We first explored replacing V by Fe. This was done because it has previously been employed as an element of low radioactivity (for nuclear application requiring low-activation materials). The calculations are shown in the supplementary material. However, the addition of Fe in both low and high elemental concentrations promotes the formation of several undesired intermetallic phases instead of single-phase solid-solution, as well as lowering the melting point of the system. Due to failure of this approach, a quinary HEA was planned considering the principles introduced by the VEC model. It has been shown that BCC RHEAs can exhibit high-stability and suitable mechanical properties when the VEC value remains below 4.4[24,25]. The strategy then consists in finding a combination of elements that can lower the VEC value. It is worth emphasizing that a VEC value of 4.4 is based on a few works on HEAs and therefore is not considered as an absolute number. Group IV elements can then be used to pursue this approach (e.g., Ti, Zr, and Hf) and modify the W-Ta-Cr-V RHEA. Due to its high melting point, Hf was firstly selected with a minimal concentration to avoid a material with high activation under in-reactor conditions and to maximize the probability of obtaining a single-phase HEA (all calculations including CALPHAD for selected compositions are in the supplementary material). In this case, 5.2 was shown to be the minimum achievable VEC value, and certain compositions were shown to favor the formation of a single-phase BCC HEA.

As mentioned in the methods section, two films were prepared: (1) 100 nm thin film to study the thermal stability and the irradiation response to He implantation and dual-beam irradiation, and (2) a 3 μm film for morphology and hardness analysis via APT and nanoindentation, respectively. The deposited thin and thick (bulk) films with Hf had compositions of $W_{29.4}Ta_{42}Cr_{5.0}V_{16.1}Hf_{7.5}$ and $W_{31}Ta_{34}Cr_{5.0}V_{27}Hf_{3.0}$ which are predicted by CALPHAD to form a single-phase BCC structure

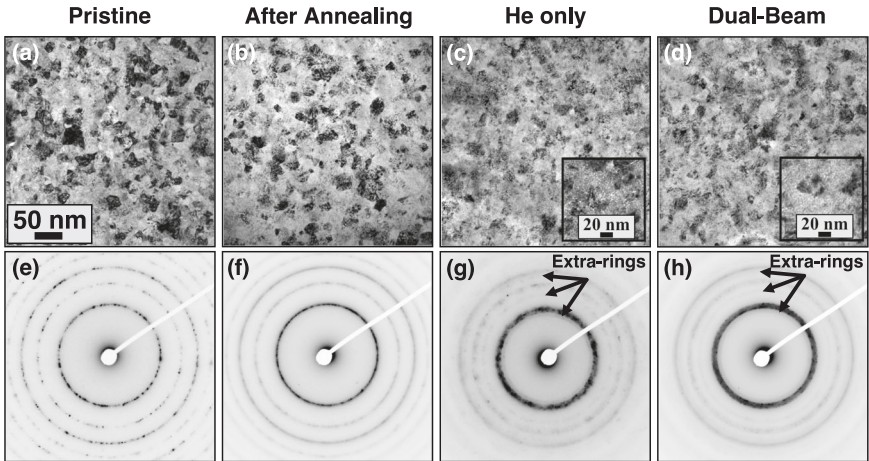

**Fig. 1 | Sample morphology under different severe environments.** Bright-field TEM (BF-TEM) images showing the nanocrystalline microstructure of the **a** as-deposited, **b** annealed, **c** single-beam He implanted, and **d** dual-beam irradiated $W_{29.4}Ta_{42}Cr_{5.0}V_{16.1}Hf_{7.5}$ 100 nm film HEA. The insets in **c**, **d** show underfocused micrographs featuring bubbles. The corresponding diffraction patterns are shown in **e**–**h**. Note: scale bar in **a** applies to **b**–**d**.

over a wide temperature range (illustrated in the supplementary). We acknowledge that both films have different compositions, which is due to challenges in depositing specific compositions from five different targets within this quinary system. However, it should be noted that different compositions, tested in thin film forms, led to similar single-phase BCC microstructures with similar irradiation responses (in terms of dislocation loop and cavity formation, as well as grain size stability). Only one composition is discussed throughout this paper for clarity. Hence, the thick film composition follows similar irradiation resistance. The morphology of the as-deposited, and after in situ TEM annealing of the thin film at 1173 K are shown in Fig. 1a, b, respectively. A single-phase RHEA is clearly shown in the corresponding diffraction patterns. Two types of ion irradiations were then performed: (1) 16 keV He implantation to compare with other W-based materials and to maximize swelling and (2) dual-beam (1 MeV Kr$^+$ + 16 keV He$^+$) to test the materials under reactor-relevant conditions where heavy ions mimic fast neutron damage, albeit with a different energy spectrum, and the implanted He mimics the gas production from transmutation reactions. Previous work by El Atwani et al. on W under sequential and simultaneous dual-beam conditions showed that simultaneous dual-beam leads to different loop and cavity damage evolution, compared to single or sequential beams[30]. The amount of He/dpa, however, was kept very high in an effort to evaluate the quinary RHEA response to extreme conditions.

TEM micrographs from post-implantation and after dual-beam ion irradiation samples are shown in Fig. 1c, d, respectively. Thickening of the BCC rings as shown in Fig. 1g, h indicates the presence of lattice strain, possibly due to defect formation and concentration gradients. It is also important to note that additional rings are present after irradiation. Although these rings were not identified, they are anticipated to correspond to a different phase with a very low phase fraction given the low-contrast intensity exhibited by the rings (e.g., shallow surface oxides due to irradiation and/or high-temperature exposure). Experimental and modeling results (discussed below) indicate no metallic segregations in the grain matrices. Figure 2 shows the corresponding chemical distribution for the samples. Careful analysis of the elemental maps revealed that after annealing, Hf segregates to grain boundaries, and V and Cr are also inhomogeneously distributed throughout the sample, thus indicating segregation. After irradiation, Hf starts to deplete from the grain boundaries (Hf concentration decreases compared to the annealed sample).

The thermal stability in terms of grain size is studied via in situ TEM annealing and irradiation and the results are plotted in Fig. 3. The average grain size of the as-deposited sample was 24.5 ± 1.3 nm. After

annealing to 1173 K for ~30 mins in the TEM, the grain size increased to 26.3 ± 1.1 nm. After He implantation, the grain size further increased to 36.3 ± 1.8 nm. However, during the dual-beam irradiation, grain refinement occurred, and the average grain size dropped to 21.1 ± 1.4 nm after a total irradiation time of ~94 mins. The overall material stability in terms of phases and segregation is discussed further below.

The irradiation resistance of the alloy is also studied in terms of both dislocation loop formation and cavity formation. Dislocation loops were not detected even after 8.5 dpa and at 9.13% He implantation during the dual-beam experiment (Supplementary Movie 1 is attached to the supplementary material). Only cavities were observed. Quantification of the cavity density, average cavity size, and total change in sample volume is shown in Fig. 4. Cavities were visible in the microscope after 5 dpa irradiation in the case of dual-beam and ~8.5% He implantation. No increasing trends in cavity volume, density, or change in volume occur after that, and saturation is evident for these dpa/He percentage ranges.

Nanoindentation was performed before and after annealing and after irradiation on the thick deposited specimen to shed light on possible changes in the alloy hardness. This specimen had a composition of $W_{31}Ta_{34}Cr_{5.0}V_{27}Hf_{3.0}$ and was irradiated with 400 keV Ar$^{+2}$ ions at 1073 K to 10 dpa. The results are shown in Fig. 5. The average hardness of the unirradiated sample was measured to be 13.25 GPa. After annealing the sample hardness increased to 16.25 GPa, while after irradiation the sample hardness reached a value of 20 GPa. The unirradiated sample is ~75% harder than pure W[31]. The change in hardness after both annealing and irradiation will be discussed based on morphology and further analysis performed and discussed below.

To predict the morphology of the samples, first-principal calculations of phase stability and chemical short-range ordering in the $W_{29.4}Ta_{42}Cr_{5.0}V_{16.1}Hf_{7.5}$ (thin film) and $W_{31}Ta_{34}Cr_{5.0}V_{27}Hf_{3.0}$ (thick film) as a function of temperature were performed. The chemical short-range order parameters (SRO) as a function of temperature as calculated from a 16,000 atom simulation cell are plotted in Figs. 6 and 7, respectively, to discuss the SRO effect on the material morphology. The two compositions demonstrate different first-order transitions and ODTT, which reflects on the distribution of precipitates in the alloys at various temperatures. Among the considered pairs of atoms, the strongest attraction is observed for the Cr-Hf pair, which presents the most negative SRO parameter at low temperatures for both RHEAs. However, the temperatures at which the chemical order between Cr and Hf atoms vanishes are notably different, namely 1080 K for $W_{29.4}Ta_{42}Cr_{5.0}V_{16.1}Hf_{7.5}$ and 620 K for $W_{31}Ta_{34}Cr_{5.0}V_{27}Hf_{3.0}$ alloy (see

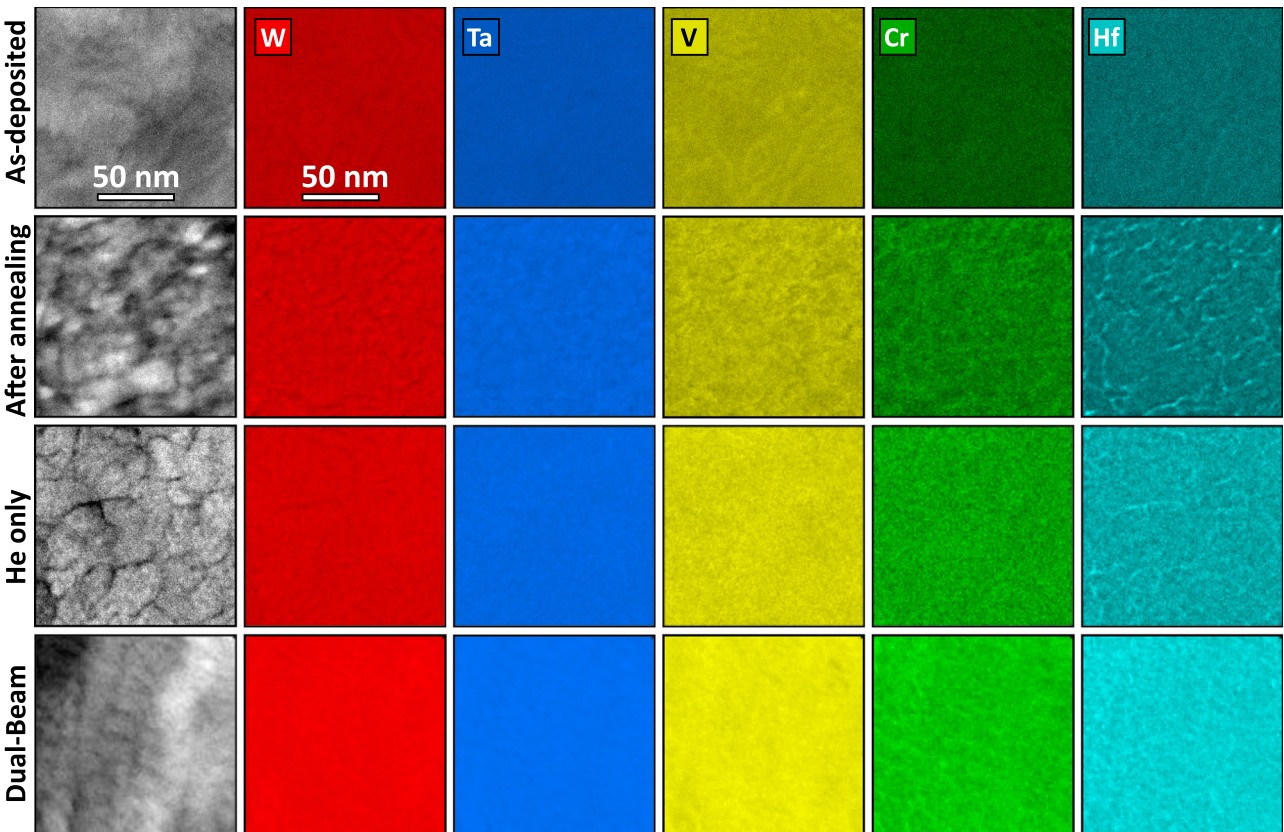

**Fig. 2 | Elemental mapping of the samples exposed to different severe environments.** EDX chemical comparison via TEM of the as-deposited, annealed, He implanted, and dual-beam irradiated $W_{29.4}Ta_{42}Cr_{5.0}V_{16.1}Hf_{7.5}$ 100 nm film HEA. Note: presented scale bar apply to all micrographs in the figure.

Fig. 6). Consequently, the Cr-Hf precipitate is visible in the simulation cell of $W_{31}Ta_{34}Cr_{5.0}V_{27}Hf_{3.0}$ alloy in Fig. 7a only at 300 K, whereas in the case of the alloy with higher Hf concentration, it is observed in the three chosen simulation cells up to 1000 K (see Fig. 7b).

## Discussion

### High thermal stability

The synthesized quinary W-Ta-Cr-V-Hf RHEA shows high thermal stability in terms of grain coarsening. Even though there was an increase in the grain size during He implantation, the material preserved its nano-crystallinity and showed only approximately 10 nm change in grain size after ~8.45 at.% He implantation. The previously studied quaternary W-Ta-Cr-V RHEA demonstrated a similar thermal stability regarding grain morphology[22]. One of the properties that distinguishes this quinary W-Ta-Cr-V-Hf RHEA from the previously studied RHEAs is the grain refinement observed during dual-beam irradiation. Grain refinement was observed before in other materials[32]. Two main suggested mechanisms are discussed in the literature: (i) defect clusters produced during irradiation can migrate to sub-grain boundaries and form cell structures that lead to small grain formation[32] and (ii) cascades that are larger than the grain size can form stacking faults across the grain breaking it into two separate crystalline structures (observed in FCC material)[33]. Overlapping cascades that are smaller than grain size are also believed to potentially cause grain refinement[32]. The results here demonstrate that grain refinement only occurred when the heavy ions are introduced. Even with 16 keV He ions, displacement should occur, and defects are then generated via Frenkel-pair production. However, grain refinement was not observed for the He only case, but rather grain growth occurred as shown in Fig. 3. This suggests that such a grain refinement observed in the dual-beam irradiation case was caused by a cascade effect. The in situ nature of the experiments performed in this work has allowed us to observe the grain

refinement effect in real-time, albeit the division of some grains into sub-grains was observed at discrete time intervals since the fragmentation process was occurring at faster time scales than the frame rate used during the data recording. Figure 8 shows grain fragmentation that occurred during the dual-beam irradiation. Molecular dynamics simulations have shown that high-energy (50 keV) recoil events begin to divide in sub-cascades[34,35], each with a diameter on the order of 10 nm. This upper limit on cascade size is near the initial grain-size of the W-Ta-Cr-V-Hf RHEA, allowing fragmentation to take place whereas, in coarser-grained materials, a similar event would likely result in defect migration to boundaries. As He implantation only causes grain growth and dual-beam irradiation leads to grain refinement, it might be expected that single-beam heavy ion irradiations result in further grain refinement. Therefore, if used in a nuclear environment where cascades are expected due to highly energetic neutrons, this material is expected to keep its structural integrity in terms of grain size.

Another significant improvement of this material system over the previous quaternary W-Ta-Cr-V RHEA, is its stability in terms of phase separation or segregation. The EDX demonstrated depletion of Hf from grain boundaries during irradiation which can be attributed to the Inverse Kirkendall effect (IKE)[36] and/or ballistic mixing during irradiation[37]. To demonstrate further the microstructural stability of this material, APT (Fig. 9) was performed on thick films of the as-deposited sample and the ex situ single-beam Ar implanted $W_{31}Ta_{34}Cr_5V_{27}Hf_3$ as it was extremely challenging to perform APT on the in situ irradiated thin films (~100 nm). The APT results from the as-deposited sample demonstrated compositional striations that were not detected by EDX (which was also the case for the W-Ta-Cr-V RHEA)[22]. The striations contained two main layers, a W-Ta layer, and a Cr-V-Hf layer. The APT needle extracted from the irradiated sample contained two parts: deep unirradiated, but heated to 1073 K and an irradiated part at 1073 K. The compositional striations were shown to

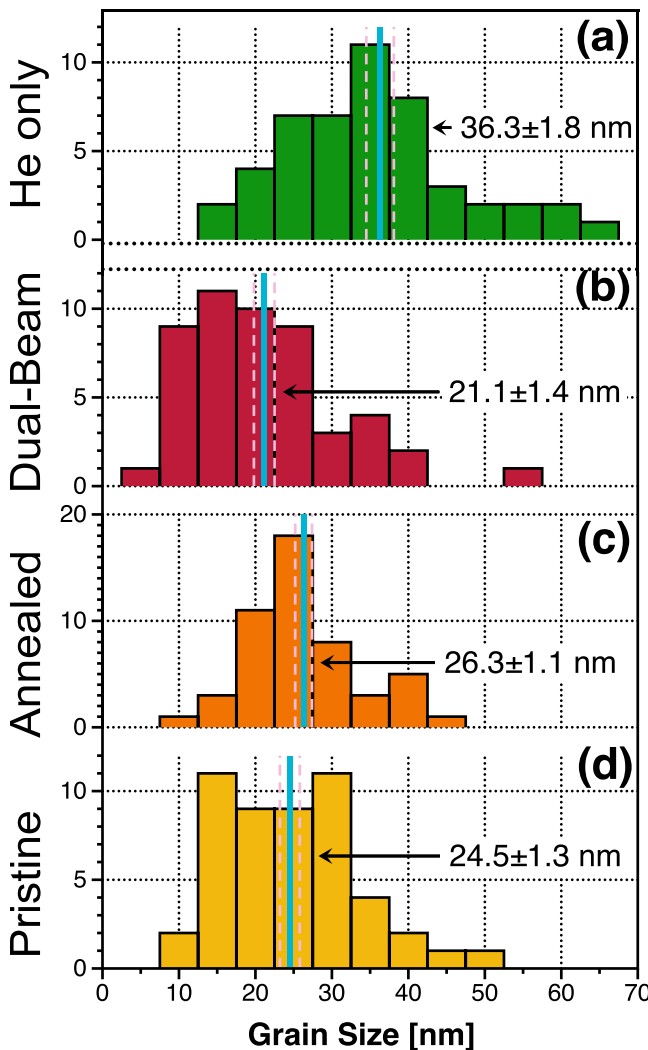

**Fig. 3 | Grain growth or refinement after implantation or irradiation.** Quantification via BF-TEM micrographs of the grain-size distribution of the **a** as-deposited, **b** annealed, **c** dual-beam irradiated, and **d** He implanted $W_{29.4}Ta_{42}Cr_{5.0}V_{16.1}Hf_{7.5}$ 100 nm alloy. Note: the blue line indicates the average value and the dashed lines denote the standard error of mean.

persist in the heated and unirradiated part, but Hf and Cr have lost their correlation which is predicted by the SRO first-order transition (Fig. 6). The irradiated part showed homogenization of the elements and no precipitation. This is unlike the W-Ta-Cr-V RHEA where irradiation led to uniform and dense Cr-V precipitation[22]. This can be discussed based on the SRO as a function of temperature (Fig. 6) obtained through the modeling analysis. A negative SRO indicates ordering and a tendency to form intermetallic compounds, while a positive SRO indicates tendency for phase separation. Homogenization of the elements occurs when all SRO values tend to zero which marks the ODTT of the alloy. The $W_{31}Ta_{34}Cr_5V_{27}Hf_{3.0}$ showed an ODTT of ~1120 K which roughly coincides with the irradiation temperature. Considering the irradiation-induced ballistic mixing and the corresponding atomic displacement and ordering[37], homogenization can occur at lower temperatures. Therefore, we expect to see homogenized chemical distribution of the material after irradiation which was the case in this work. For the in situ irradiated samples, TEM images, and EDX showed no spherical precipitates and for that composition ($W_{29.4}Ta_{42}Cr_{5.0}V_{16.1}Hf_{7.5}$), the SRO and the corresponding atomic configurations (Figs. 6 and 7) showed a slightly smaller (1080 K) ODTT (but rather different behavior at lower temperatures) and,

therefore, no precipitation is expected during irradiation at 1173 K. It should be stated that the SRO and atomic configurations of the W-Ta-Cr-V RHEA predicted both density and size of the Cr-V precipitation[22]. We then conclude that: (1) no precipitation is expected in the studied compositions at temperatures close to or over the ODTT, (2) changing composition can modify SRO and the material performance as a function of temperature and (3) experiments and modeling agree, thus allowing better understanding and prediction of the material's morphology as a function of temperature, which would constitute a design methodology for developing RHEAs/HEAs for nuclear applications with different temperature requirements.

## High tolerance to energetic particle irradiation

The ion irradiation response of this alloy is characterized by: (1) no dislocation loop formation, (2) negligible change in overall volume due to bubble formation, (3) no preferential bubble formation on grain boundaries despite the high atomic percentage of He implantation at 1173 K, (4) thermally stable grain-size (with 10 nm increase during He implantation up to 8.45% and grain refinement under dual-beam ion irradiation), and (5) no precipitation when used at reactor-relevant temperature (due to intermediate ODTT values). Pure W, for example, suffers from the formation of large facetted bubbles on the grain matrices and preferential bubble formation with larger sizes on the grain boundaries when implanted with similar He energy but at an even lower implantation value (6.3%)[38]. At 1173 K, He-vacancy complexes are mobile in W[39] and can coalesce to reach a change in volume of ~1.7% from the grain matrices contribution only. The grain boundaries, with much larger bubbles (~200 nm$^2$ compared to ~25 nm$^2$ in the grain matrices) had a larger contribution of ~7.4% change in volume[38]. Nanocrystalline W and ultrafine W-TiC alloys implanted with 2 keV He$^+$ at 1223 K to one order of magnitude lower fluence show 0.4% and 0.6% change in volume from the grain matrices respectively, and while preferential He bubble formation on the grain boundaries still occurred, its contribution for the quantification of change in volume was very challenging[40]. Under dual-beam irradiation, W demonstrated dislocation loop and dislocation network formation of high density and size (~100 nm$^2$ loop size and $0.5 \times 10^{-3}$ nm$^{-2}$ density) when the total dose was only 0.25 dpa. The total change in volume was ~0.65% (from grain matrices only)[30]. The current alloy presents no loop formation and the total change in volume is ~0.3 % after 8.45 dpa and higher percentage of implanted He. Additionally, no preferential large and facetted cavity formation at the grain boundaries was observed. It should be noted that possible surface effects are expected to diminish when the grain boundary-to-surface ratio approaches the value of 1[41]. In this HEA, the surface ratio is approximately 10. Furthermore, surface effects in the RHEA system are expected to be smaller than in pure W due to the rougher defect migration landscapes in HEAs[42].

The irradiation response to loop and cavity formation in W-based RHEAs was attributed to high interstitial-vacancy recombination[22], which was also implied by Zhao who demonstrated via DFT calculations that vacancies in W-Ta-Cr-V RHEA present smaller migration energies compared to pure W and overlapping interstitial and vacancy formation energies[43]. It is also shown by Zhao that most interstitial dumbbells are along the [110] direction[43], which in BCC systems, involves a sequence of rotational and translational jumps as described by Schilling[44]. Others related the high irradiation resistance of HEAs to lattice distortion and sluggish diffusion effects[45] or the difficulty of defect clustering[46]. Using electronic structure calculations as implemented in VASP code, formation, and migration energies of He were computed in the W-Ta-Cr-V system[23]. The He average migration energy was found to be 0.156 eV in the alloy compared to 0.06 to 0.081 eV in pure W and the He formation energy was approximately 2 times lower[47,48]. This rough energy landscape implies that He has a higher tendency

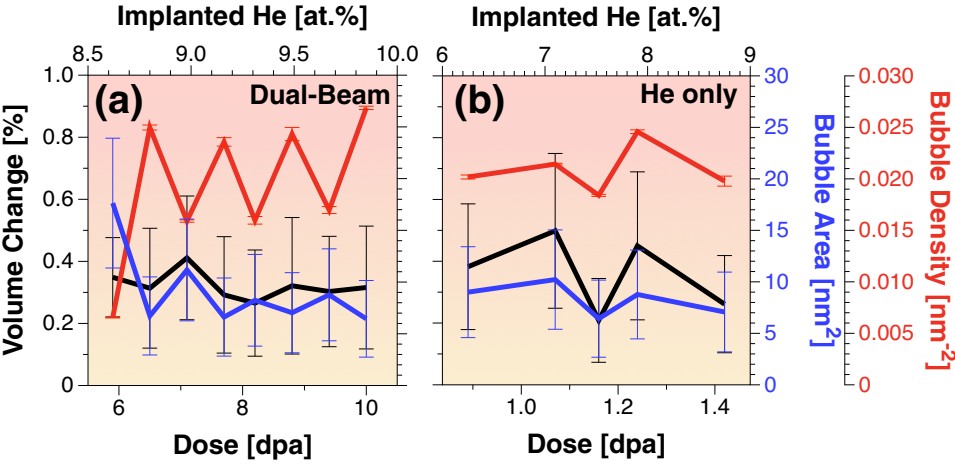

**Fig. 4 | Assessment of irradiation damage.** Plots showing the average volume change, average bubble area, average areal number density of the dual-beam irradiated **a** and He implanted alloy **b** as a function of dpa and percentage of implanted He in the $W_{29.4}Ta_{42}Cr_{5.0}V_{16.1}Hf_{7.5}$ 100 nm alloy. The scale bars denote the standard error of the mean.

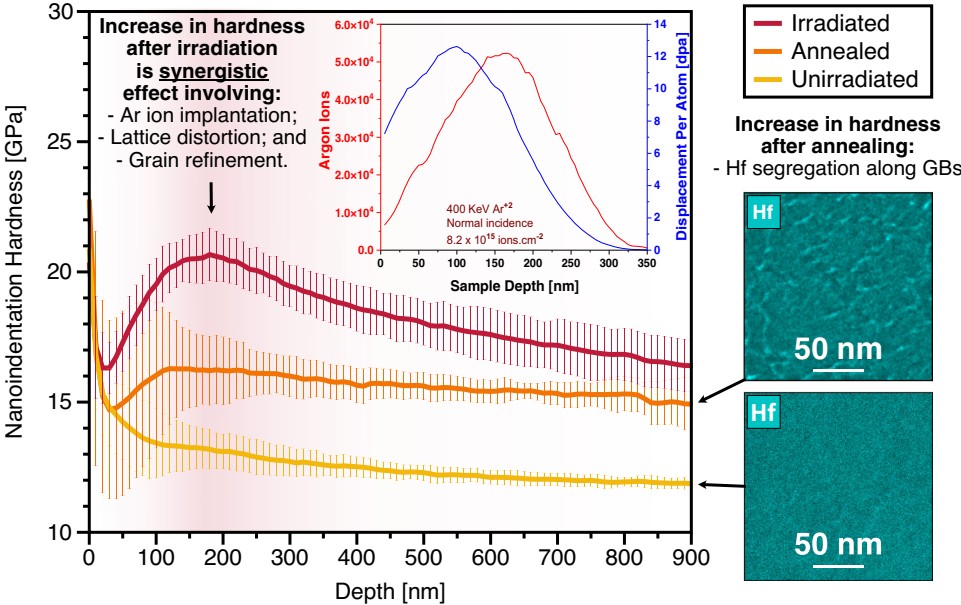

**Fig. 5 | Assessment of strength and hardening.** Nanoindentation results showing hardness vs depth of the $W_{31}Ta_{34}Cr_{5.0}V_{27}Hf_{3.0}$ thick HEA film before and after annealing, and post-irradiation to 10 dpa. Increase in hardness after annealing is solely associated with Hf segregation along the GBs whereas the increase after irradiation is a synergistic combination between Ar implantation, lattice distortion caused by irradiation ballistic mixing, and grain refinement. Error bars denote the standard deviation.

to quickly find a fairly stable site that can act as a bubble nuclei, and also to bind to other slowly migrating He interstitial atoms. The clustering of He will slow them further down and a trend for smaller and lesser bubble nucleation will follow, and therefore, uniform bubble distribution with no preferential bubble formation at the grain boundaries or a wide distribution of bubble sizes occur. The absence of preferential bubble formation at grain boundaries can also stem from the high migration barrier of He-vacancy complexes, which still needs to be further investigated. Grain boundaries can also play a role in the irradiation resistance of the alloy acting as defect sinks. The contribution of grain boundaries, however, compared to grain matrices to the irradiation resistance has been studied comparing NC HEA and coarse grain HEA to NC-W and coarse grain W and it was found that the dominant factor is the grain matrix chemistry[49].

## Changes in nanoindentation hardness

The radiation resistance of the alloy morphology must be reflected on mechanical properties. Nanoindentation hardness measurements were performed for this purpose on one composition of this alloy system. The as-deposited sample had a hardness of 13.25 GPa.

The increase in hardness after annealing was ~3 GPa and this can be directly attributed to Hf segregation to grain boundaries as revealed in Fig. 2 and in the STEM-EDX mapping as insets in Fig. 5.

After irradiation, the hardness further increased by 3.75 GPa. This RHEA possesses high irradiation resistance and shows no dislocation loop formation, even under dual-beam conditions (where He can bind to vacancies allowing interstitials to coalesce faster). Small defect clusters invisible in the TEM can affect hardness. However, considering the dispersed barrier hardening (DBH) model[50] and other experimental work in W[51], the increase in hardness in this RHEA cannot at all

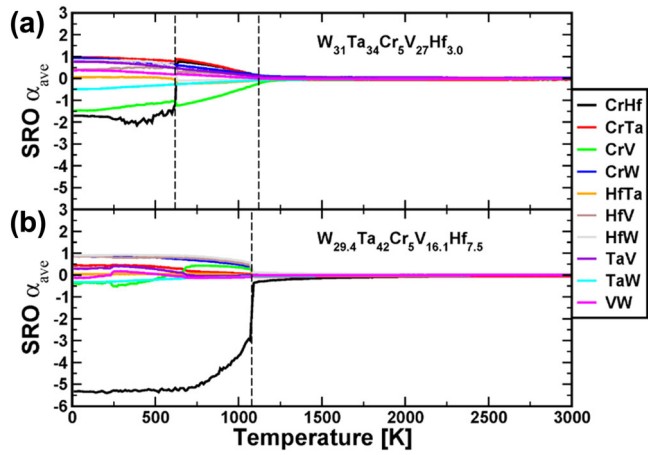

**Fig. 6 | Chemical short-range ordering and order to disorder transition temperatures.** Chemical short-range order for different pairs of atoms as a function of temperature for both the **a** $W_{31}Ta_{34}Cr_{5.0}V_{27}Hf_{3.0}$ thick film used for mechanical properties and APT studies (top panel) and **b** $W_{29.4}Ta_{42}Cr_5V_{16.1}Hf_{7.5}$ thin film used for the in situ experiments (bottom panel) alloy.

be justified by such invisible defects. Two other factors can affect the hardness of the irradiated RHEA. First, some change in hardness could be a result of material homogenization which occurred due to the transition from ordered to disordered state, as discussed above, and which enhances lattice distortion[52]. Lee et al.[53] demonstrated improved mechanical strength in homogenized NbTaTiV HEA which were shown to stem from lattice distortion during deformation. Others have also suggested lattice distortion to be the dominant factor affecting the mechanical properties of HEAs[54]. Another factor that can cause a change in hardness in this HEA is the refinement of grain size when irradiated with heavy ion as demonstrated and illustrated earlier. As the grain size in this RHEA is very small (Fig. 3) and hardness follows a power law with grain size, a small refinement of grain size could result in a large change in hardness. This change is higher when the grain size is very small and approaches the edge of the transition from the Hall-Petch to the Inverse Hall-Petch effect[55]. In Fig. 5, the irradiated curve approached the annealed curve at high depths that exceed the irradiation depth by a factor of ~3, suggesting that the change in hardness (~1.6 GPa) at that depth is due to elemental segregation to grain boundaries after annealing. The 3.75 GPa difference at 200 nm depth can also be explained by Ar implantation as calculated per SRIM, shown in the inset plot of Fig. 5: the hardness curve after irradiation follows the peak of Ar implantation at depths ~200 nm.

One can note that the deconvolution of multiple effects contributing to such a hardness increase after irradiation depends on multiple synergistic processes. Observing the results presented in Fig. 5, we conclude that at the peak of the Ar irradiation/implantation profile (~200 nm), the hardness increase is a result of a synergistic combination of Ar implantation, lattice distortion, and grain refinement, thus not solely due to one specific cause. This fact is confirmed when further depths—beyond the zone of influence of Ar irradiation/implantation—are analyzed. Beyond the irradiation profile, the difference between the irradiation and annealing curves monotonically decreases. At a depth around of 900 nm, such a difference is within the error bars. This small difference can be attributed to the extended zone of plastic deformation after irradiation, which is around 3 times the depth of the irradiation peak[56,57].

The hardness increase in both annealing and irradiation cases are herein elucidated by experiments. Nevertheless, it is expected that this work will promote further studies on understanding this complex mechanical response of RHEAs when exposed to extreme irradiation conditions. It also should be noted that different compositions of this

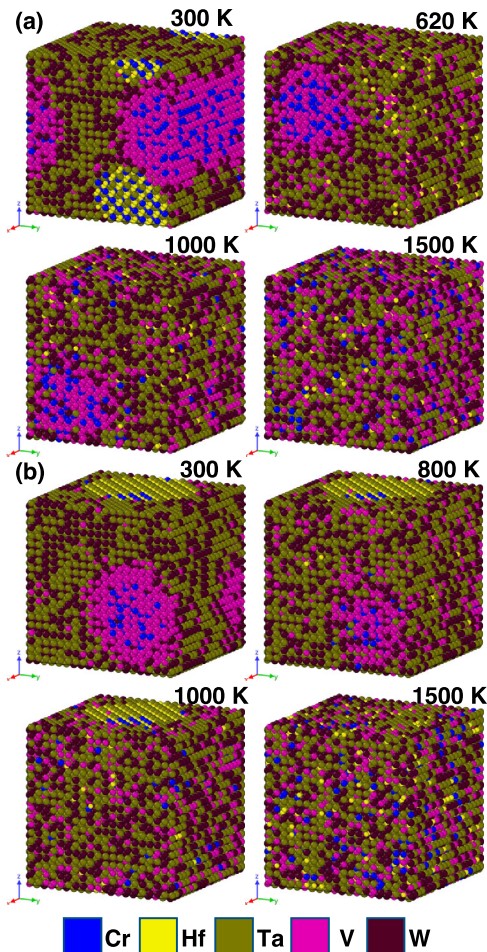

**Fig. 7 | Atomic distribution of different HEA compositions.** 16,000 atom simulation cells as a function of temperature of the **a** $W_{31}Ta_{34}Cr_{5.0}V_{27}Hf_{3.0}$ (thick film) and **b** $W_{29.4}Ta_{42.0}V_{16.1}Cr_{5.0}Hf_{7.5}$ (thin film) alloys. Temperatures were chosen to reflect the results from Fig. 6.

alloy system can have different SRO parameters and different elemental segregation behavior (Figs. 6 and 7), and therefore, are expected to have different mechanical properties. Herein, we have focused on one composition ($W_{31}Ta_{34}Cr_{5.0}V_{27}Hf_3$). The effect of SRO on the mechanical behavior of future RHEA in extreme environments is another research topic to be further explored.

In summary, a design methodology, based on thermodynamic calculations combined with experimental and simulation components, has been herein established for the development of promising irradiation-resistant RHEAs. Following this methodology, a quinary RHEA was designed and studied in terms of irradiation resistance, thermal stability, and strength. Such HEA shows irradiation resistance to dislocation loop and cavity formation under heavy ion irradiation and He implantation (even after 9% He implantation) with hardness values (in the annealed state) that are higher than the value in pure W. The irradiation resistance to loop and cavity formation under dual-beam and He implantation is higher than in other HEAs and pure metals studied in the literature. All results are discussed correlating the experimental results with modeling insights. We acknowledge that further work (e.g., response to neutron irradiation) is necessary to advance the technology readiness level (TRL) of this material and that the effect of composition changes in mechanical strength should be investigated. Tensile measurements should follow on coarse grain and bulk forms. Insights provided by the agreement between experimental and modeling results in this work can be used to design other alloys for different applications.

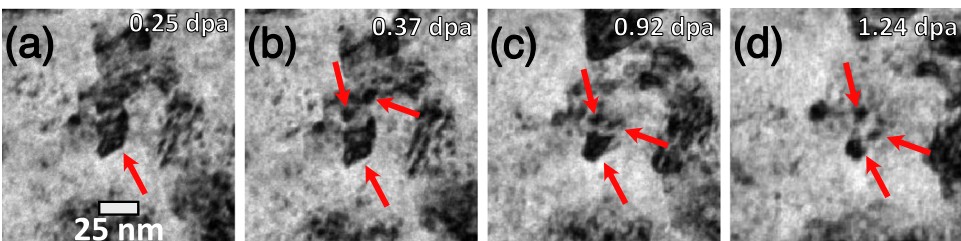

**Fig. 8 | Evidence of grain refinement during irradiation.** BF-TEM snapshots from the in situ irradiation TEM video tracking the cascade-induced grain fragmentation during irradiation: **a** 0.25 dpa, **b** 0.37 dpa, **c** 0.92 dpa, and **d** 1.24 dpa. Note: scale bar in **a** applies to all micrographs in the figure.

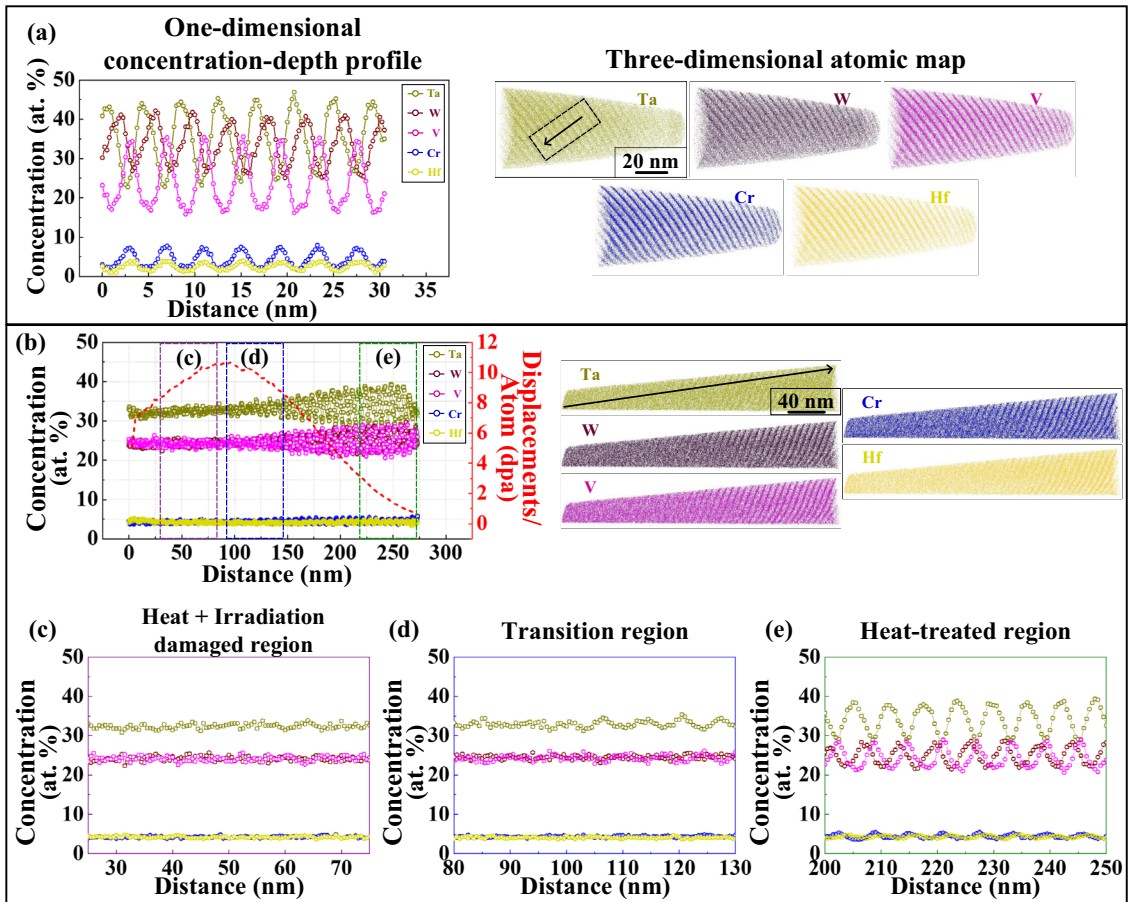

**Fig. 9 | 3D atomic maps and concentration profiles via atomic probe tomography. a** APT atom maps of the as-deposited condition and the corresponding 1D concentration profile. **b** APT atom maps of the ex situ heavy ion irradiated thick film alloy ($W_{31}Ta_{34}Cr_{5.0}V_{27}Hf_3$) and the corresponding 1D concentration profile (left axis) the and dpa vs. depth profile (right axis). **c**–**e** 1D chemical profiles showing the transition from high dpa to low dpa regions.

## Methods

### Fabrication of the W-Ta-Cr-V-Hf alloy

The material was synthesized using magnetron-sputtering deposition from metal targets of 99.99% purity using power of 200, 400, 50, 350, 25 Watts for W, Ta, Cr, V, and Hf, respectively. The deposition was performed at room temperature and 3 mTorr pressure at no bias voltage. Two sets of depositions were performed: (1) 100 nm thin film on NaCl substrate and (2) 3 μm film on W substrate. Transmission electron-microscopy (TEM) samples were then prepared by floating the film on a standard molybdenum TEM grid using 1:1 ethanol/water solution. Nanoindentation was performed on the 3 μm film on W substrate. The thin (for in situ TEM) and the thick films (for mechanical properties and atomic probe tomography, APT, studies) had compositions $W_{29.4}Ta_{42}Cr_{5.0}V_{16.1}Hf_{7.5}$ and $W_{31}Ta_{34}Cr_{5.0}V_{27}Hf_{3.0}$, respectively.

### Pre-irradiation characterization of the thin film TEM samples

Before ion irradiation, the films were analyzed using energy dispersive X-ray (EDX) spectroscopy and selected area diffraction (SAED) in an FEI Titan 80–300 TEM operated at 300 keV. The material was then annealed in situ within the TEM at 1173 K for 10 minutes. EDX and SAED were performed on the thin film sample in the as-deposited and post annealed, implanted, and irradiated conditions.

### In situ TEM ion irradiation of the thin-film samples

The RHEA material was irradiated in situ at the Intermediate Voltage Electron Microscope (IVEM)-Tandem Facility at Argonne National Laboratory with 1-MeV $Kr^{+2}$ and 16 keV $He^+$. Two irradiation conditions were performed: (1) dual-beam irradiation with 1-MeV $Kr^{+2}$ and 16 keV $He^+$ and (2) single-beam implantation with 16 keV $He^+$. In the dual-beam ion irradiations, the material was irradiated to 8.5

displacements per atom (dpa) and ~9% He. The implanted He per dpa ratio was ~1.07% He/dpa. The corresponding Kr and He fluences were $2.74 \times 10^{15}$ and $5.84 \times 10^{16}$ ions/cm$^2$ with average fluxes of $4.86 \times 10^{11}$ and $1.04 \times 10^{13}$ ions/cm$^2$/s, respectively. In the single-beam implantation, the material was implanted with He up to ~9%, same amount as in the dual-beam irradiations. The dpa and He implantation profiles were calculated using the Kinchin-Pease model in the Stopping & Range of Ions in Matter (SRIM) Monte Carlo computer simulation code (version 2013)[58] and 40 eV[59] was taken as the displacement threshold energy for all elements. The dpa and He implantation profiles are presented in the supplementary material. The irradiation temperature in both dual and single-beam irradiation experiments was set to 1173 K. In situ videos were collected for irradiation-induced damage quantification. The damage quantification procedure is well described in ref. [60]. For bubble density determination, random arrays of 15 nm circles were drawn on each grain using ImageJ software[61] and bubbles are counted in every circle and divided by the area of the circle. An average is taken as the bubble density. For bubble area determination, ImageJ software was used to draw a circle around the perimeters of the bubbles on every grain. The areas of the circles are determined, and an average is taken. The overall change in volume after irradiation was calculated using $\Delta v/v = \frac{4}{3}\pi r_c^3 N_v$ where $N_v$ is the bubble density in a 100 nm thick foil and $r_c$ is the radius of the bubble.

## Ex situ ion irradiation of the thick film samples

The ex situ ion irradiations were performed with 400 keV Ar$^{+2}$ ions on a 200 kV Danfysik Research Ion Implanter at the Ion Beam Materials Laboratory at Los Alamos National Laboratory. The beam flux and the fluence were $1.8 \times 10^{12}$ ions/cm$^2$/s and $8.2 \times 10^{15}$ ions/cm$^2$ (10 dpa), respectively. The sample was mounted on a heating stage with silver paste and the stage temperature was kept at 1073 K during the irradiation and monitored continuously with a thermocouple mechanically attached to the heating stage. The ex situ irradiation beam parameters were chosen to match the damage process of the in situ irradiation conditions at IVEM for subsequent atom probe tomography (APT) analysis. The dpa and He implantation profiles are presented in the supplementary material.

## Post-irradiation characterization of the thin film samples

SAED and high-resolution EDX measurements were performed after the irradiation. APT was performed on the as-received, after-annealing, and after-irradiation conditions. CAMECA's Integrated Visualization and Analysis Software (IVAS) was utilized to reconstruct and analyze the APT data. APT samples were fabricated using standard lift-out and sharpening methods as described by Thompson et al. [62]. Briefly, wedges were lifted out, mounted on Si microtip array posts, sharpened using a 30 kV Ga+ ion beam, and cleaned using a 2 kV Ga+ ion beam. For the irradiated samples, the top of the needle was located as close to the surface as possible (<50 nm from the surface). The APT experiment was run using a CAMECA LEAP 4000XHR in laser mode with a 30 K base temperature, 80–100 pJ laser energy, a 0.5% detection rate, and a pulse repetition rate set to capture all elements in the mass spectra.

## Mechanical properties assessment of the thick film samples

Nanoindentation tests were performed on the as-deposited and post-irradiated materials using a Keysight G200 Nanoindenter with a diamond, pyramidal (Berkovich) tip to a final displacement of 1000 nm with a constant strain rate (loading rate divided by the load) of 0.05 s$^{-1}$. Continuous stiffness measurements (CSM) were performed at a frequency of 45 Hz and 2 nm displacement amplitude. Since the irradiated layer is 200–300 nm and size effect can be around three times the indented depth (for brittle materials), the hardness and modulus measurements were obtained at 200 nm indentation depth.

## Modeling techniques

**Density-functional theory.** Density-functional theory (DFT) calculations were performed using the Vienna Ab Initio Simulation Package (VASP)[63]. For the exchange and correlation, a generalized gradient approximation of the Perdew-Burke-Ernzerhof form (GGA-PBE), with projector augmented wave method was used. Both the semi-core p electron and magnetism were not included in this study since their contribution does not significantly affect the calculations. The convergence criteria for energy were set to 10$^{-5}$ eV per cell. The cells were relaxed to a force convergence criterion of 10$^{-3}$ eV Å$^{-1}$. The Monkhorst-Pack mesh spacing[64] was such that it corresponded to a $14 \times 14 \times 14$ k-point mesh of a two-atom BCC cell. The plane-wave cutoff energy used was 400 eV.

The Alloy Theoretic Automated Toolkit (ATAT) package[65] was used to generate a cluster expansion (CE)-based configurational energy expression fitted to DFT energies. A modified database of 58 initial structures for each binary subsystem was used following D. Nguyen-Manh et al. [66]. For ternary subsystems, 94 ternary structures were constructed from initial binary structures by replacing the atoms in one of the nonequivalent positions from the symmetry point of view with the third type of atom[67]. For quaternary and quinary structures, a database from ref. [68] was used and modified for the W-Ta-Cr-V-Hf system. In total, 1511 BCC structures including quinary, binaries, ternaries, and quaternaries subsystems of the W-Ta-Cr-V-Hf HEA were used in the fitting. During DFT relaxation, the volume and shape of the cell were allowed to change. Only structures without large distortions compared to the starting configuration were used in the fitting. From ATAT's toolkit package, the *checkrelax* function was used to ensure the square-root sum of each element of the strain tensor squared no larger than 0.1. Additionally, the common neighbor analysis algorithm in OVITO[68] was used to ensure the structure remained mainly in a BCC phase.

## Cluster expansion formalism

From DFT, the enthalpy of mixing can be computed as:

$$\Delta E_f^{\mathrm{DFT}} = \frac{E_{\mathrm{DFT}} - \sum_{m=1}^{n} N_m E_m^{\mathrm{ref}}}{N} \tag{4}$$

where $E_{\mathrm{DFT}}$ is the energy of the system as calculated from ab initio, N is the total number of atoms in the supercell, $n$ is number of components in the alloy, $N_m$ is the number of atoms of type $m$, and $E_m^{\mathrm{ref}}$ is the reference energy of atom type $m$. The reference energies, $E_m^{\mathrm{ref}}$ were calculated using the DFT methodology described above with values of $-9.51075$, $-9.77225$, $-11.8619$, $-8.94221$, and $-13.0112$ eV per atom for Cr, Hf, Ta, V, and W, respectively, in a BCC crystalline structure.

In the cluster expansion formalism, the enthalpy of mixing can be expressed using an Ising-like Hamiltonian[65],

$$\Delta E_f^{CE} = \sum_{\omega} m_{\omega} J_{\omega} \langle \Gamma_{\omega'}(\vec{\sigma}) \rangle_{\omega} \tag{5}$$

where $\vec{\sigma}$ specifies an atomic configuration in the form of configuration variables. The summation is performed over all clusters $\omega$. The clusters $\omega$ are distinct under symmetry operations of an underlying lattice, the number of equivalent clusters is obtained by multiplicity $m_{\omega}$. $J_{\omega}$ are the concentration-independent effective cluster interactions (ECIs). $\langle \Gamma_{\omega'}(\vec{\sigma}) \rangle$ are point function products of occupational variables on averaged cluster $\omega'$.

The ECIs were computed from first principles through a structural inversion method (SIM)[69]. Through SIM, one can utilize the energy corresponding to a relaxed set of structures, via DFT, to calculate the cluster functions, create a set of linear equations, and fit the ECIs. To determine the accuracy of the CE model a cross-validation (CV) score is used. CV is the square-root mean difference between ab initio energies

to those obtained by CE:

$$CV = \sqrt{\frac{1}{n}\sum_{i=1}^{n}\left(\Delta E_{f,i}^{DFT} - \Delta E_{f,i}^{CE}\right)^2} \qquad (6)$$

where $\Delta E_{f,i}^{DFT}$ is the energy of structure $i$ as calculated by DFT, and $\Delta E_{f,i}^{CE}$ the energy of the same structure predicted using CE from a least-square fit to the other $(n-1)$ structural energies.

The Warren–Cowley short-range order (SRO) parameters are used to quantify the chemical ordering between pairs of different species up to the second nearest neighbor by:

$$\alpha_n^{ij} = 1 - \frac{y_n^{ij}}{c_i c_j} \qquad (7)$$

Where $\alpha_n^{ij}$ is the chemical short-range order parameter, $y_n^{ij}$ is the probability of two atoms within $n^{\text{th}}$ neighbor shell and $c_i$ and $c_j$ are the concentrations of species $i$ and $j$, respectively. $y_n^{ij}$ is calculated through a matrix inversion of correlation functions obtained via Canonical Monte Carlo (CMC)[70].

## Data availability
Some datasets generated or analyzed during this study are included in the published article and in the supplementary information files. Other datasets are available from the corresponding author upon request.

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

## Acknowledgements

This work was supported by the U.S. Department of Energy, Office of Nuclear Energy under DOE Idaho Operations Office Contract DE-AC07-051D14517 as part of a Nuclear Science User Facilities experiment. Research in this work was performed, in part, at the Center for Integrated Nanotechnologies, and Office of Science User Facility operated for the U.S. Department of Energy (DOE) Office of Sciences by Los Alamos National Laboratory (Contract 89233218CNA000001) and Sandia National Laboratories (Contract DE-NA-0003525). O.E.A. acknowledges support from the Laboratory Directed Research and Development (LDRD) program of Los Alamos National Laboratory under the early career program project number 20210626ECR. O.E.A. and E.M. acknowledge support from the Department of Energy-Fusion Energy Science pilot program under AT2030110. M.A.T. also acknowledges support from LDRD under project number 20200689PDR2. D.N.M. and J.S.W. work has been carried out within the framework of the EUROfusion Consortium, funded by the European Union via the Euratom Research and Training Program (Grant Agreement No 101052200—EUROfusion). Views and opinions expressed are however those of the author(s) only and do not necessarily reflect those of the European Union or the European Commission. Neither the European Union nor the European Commission can be held responsible for them. D.N.M. also acknowledges funding from the RCUK Energy Program Grant No. EP/W006839/1. The work at WUT has been carried out as a part of an international project co-financed from the funds of the program of the Polish Minister of Science and Higher Education entitled "PMW" in 2019; Agreement No. 5018/H2020-Euratom/2019/2. D.N.M. and J.S.W. would like to thank the support from the high-performing computing facility MARCONI (Bologna, Italy) provided by EUROfusion. APT research was supported by the Center for Nanophase Materials Sciences (CNMS), which is a US Department of Energy, Office of Science User Facility at Oak Ridge National Laboratory. The authors would like to thank James Burns for their assistance in performing APT sample preparation and running the APT experiments. Authors acknowledge Koray Iroc for preliminary CALPHAD simulations.

## Author contributions

O.E. initiated the idea, assisted in experiments, guided the research, and wrote the final draft. H.T.V. and M.T. performed the experiments and assisted in writing the final draft. C.L. wrote the proposal for performing the APT experiments and interpreted the APT results. N.K. performed the damage quantification of the samples and video analysis. J.P. performed the APT experiments. M.L. and W.C. assisted in the in situ irradiation experiments. U.T. and E.A. performed the thermophysical parameter calculation and the CALPHAD simulations. A.A. performed the modeling results. J.W. and D.N. assisted in the modeling section and developing the cluster expansion model. E.M. assisted in all the modeling parts and in developing the research idea. Y.W. performed the ex situ irradiation experiments. K.B. performed the thin film deposition. J.G. performed the nanoindentation. A.K. assisted in the research and discussion. S.F. assisted in developing the research idea and in supervising the research. All authors contributed to the discussions of the results and the manuscript preparation.

## Competing interests

The authors declare that they have no competing interests.
