## [Peer Review File · Nature Communications]

A quinary WTaCrVHf nanocrystalline refractory high-entropy alloy withholding extreme irradiation environmentsREVIEWER COMMENTS

Reviewer #1 (Remarks to the Author):

This is work on a refractory HEA that has been designed through a combination of thermodynamic calculations together with experimental and modeling approaches for use in the harsh environments of fusion reactors. The material has been processed into thin films of two different thicknesses, and irradiation resistance, thermal stability, and strength were evaluated. The material appears to have excellent irradiation damage tolerance, and the obtained properties from one of the two thin films are correlating nicely with the simulations. The conducted work is of high quality and the study overall done nicely with the manuscript being written concisely and clearly. While as such the manuscript is overall suitable for publication in Nature Communications there are several drawbacks that need to be taken into account when a decision regarding acceptance is made.

One of the main questions is how novel this work really is? While the material appears to be a promising candidate for nuclear applications, the work is based on previous research on a WTaVCr alloy that has been published recently in Materials Today Energy and Science Advances and which has been modified through the addition of Hf based on established design principles for HEAs. Previous work on the 4-component alloy combined with the application of HEA design principles should not preclude publication; however, it is far from clear what's the purpose of processing two thin films of different thicknesses that are even in terms of composition quite different with particularly Ta, V, and Hf varying significantly (W29.4Ta42Cr5.0V16.1Hf7.5 vs. W31Ta34Cr5.0V27Hf3.0) and despite the author's comment about adding only minor additions of Hf, the final amount being comparable to Cr with 5%. It can be assumed that Cr was reduced due to the formation of precipitates – if that's the case then this does not come out though. While difficulties in preparation of two thin films with the same composition are certainly appreciated, making two thin films while then presenting results from mainly one of them is somewhat distracting – what's the point of this approach particularly if the irradiation campaigns are different? – are they even comparable?! Irradiation depth certainly plays a role not just in terms of the irradiated region but also for the mechanical characterization – CSM measurements are good, but aren't they missing or at least just crossing the irradiated depth particularly when measurements from 400-500 nm depth are taken while the irradiated layer is 200-300 nm deep? These results, together with the statement that due to the large scatter in the nanoindentation data (which is known to be a problem with this method) 'only reasonable results were shown' are highly questionable! In terms of mechanical properties, it is also highly questionable what's the rationale for the authors recommending measuring ductility at larger grain sizes? Lower strength and larger ductilities at such larger grain sizes are almost guaranteed if the material could be processed using the same conditions and everything else being identical – this is, however, likely far from easy as the authors would/should have done this otherwise. If the material can be processed to larger dimensions, it has probably different composition and different phases. Next, why did the authors choose Hf instead of the other elements they mentioned as addition to their 4-component alloy and how did they measure SRO? In contrast to one of their statements, the impact of SRO on mechanical properties is highly questionable and it appears more and more that it is actually rather limited than affecting mechanical performance. In terms of the amount of volume

change associated with irradiation and its impact on properties the authors contradict themselves at several locations in their work.

Despite these raised issues, it should be stated that the work is, overall, quite nice but the presented approach that is described as material design protocol likely overstated.

Reviewer #2 (Remarks to the Author):

This work is essentially an extension of their previous work published in Science Advances 2019. The quality of the work is high, and the results are solid. The reviewer however has some major concerns before recommending its publication.

Most importantly, the logic of the storyline is not understandable to the reviewer: so they follow the criteria to design ductile RHEAs, but they only intend to verify the ductility in a separate work? This does not make any sense, does it? How can we know for sure that the design strategy works or not? As a matter of fact, the super-high hardness almost for sure indicates the material is brittle. They also used the wording of ductility enhancement for the pending work, making a hint that the ductility might be there in bulk samples with large grain size. Of course, it cannot be said for sure that it would not happen. However, lacking any evidence of ductility, the story is certainly not self-consistent.

They have such a statement: the equilibrium structure of Hf is hexagonal-closed packed (HCP) versus BCC for the other elemental components of the alloy. Hence, only small quantities of Hf can be added in order to retain a single-phase BCC structure. This statement is not true. Indeed, the Senkov alloy, equiatomic HfNbTaTiZr, has a single-phase BCC structure and contains high quantities of Hf, Ti and Zr, all of HCP structure. Therefore, the statement cannot be justified.

They gave two compositions in thin and thick films, and they also admitted that both produced films have different compositions. However, they never gave the designed compositions or compositional range. What is indeed the product (compositions) of the design strategy, i.e., to design ductile RHEAs? There is such a statement: the He average migration energy was found to be 0.156 eV in the alloy compared to 0.06 to 0.081 eV in pure W and the formation energy was ~2 times lower. Where does this 2 times lower formation energy come from?

The hardness increases after annealing and irradiation are way too significant: 3GPa and 3.75GPa, respectively. They used compositional segregation to account for the former, and segregation, grain refinement plus lattice distortion to account for the latter. This qualitative is not convincing enough, and a quantitative analysis is necessary here.

Reviewer #3 (Remarks to the Author):

This study developed a new strategy to further improve the performance of W-based high entropy alloy via microstructural design according to the combination of valence electron concentration analysis, phase diagram calculation and density-functional theory simulation. According to the predictions, some

of the compositions were experimentally prepared and tested. Based on the new protocol, the manuscript designed a promising W-Ta-Cr-V-Hf alloy with excellent properties, including irradiation resistance, thermal stability, and strength. The agreement between experimental and modeling results provides a valid strategy for designing high performance materials for extreme environmental applications. This is a nice manuscript with sufficient novelty for publication in Nature Communications.

Some technical comments are listed below:

1. The protocol predicts a better ductility of W-Ta-Cr-V-Hf alloy. Is it possible to prepare bulk nanocrystalline alloy with the same composition? Do you expect a similar performance in the bulk nanocrystalline W-Ta-Cr-V-Hf alloy? Is it brittle?
2. Is there any limitation in current protocol? Whether is it valid for all other HEA alloys design? It is better to discuss these points in the revised manuscript.
3. Is there any surface effect in the in situ irradiation experiment of W-Ta-Cr-V-Hf alloy? For normal metals, a strong surface sink effect would influence the species and concentration of defects. It is better to explain.
3. In Fig. 5, can author explain the reason why the hardness show a turning point in the depth of 0~200 nm for the irradiated and the annealed alloys? Why is the irradiated sample harder than the annealed sample in the deeper region (400~800 nm)?
4. What is the mechanism of irradiation hardening? The authors mentioned that materials homogenization and grain refinement are two main reasons. However, a recent study shows that numerous invisible He-V clusters are the main hardening contributor in irradiated pure W (Nano Lett-2021-5798).
5. In addition, what is the mechanism of grain refinement? It is proposed that irradiation-induced grain fragmentation cause the grain refinement but the underlying mechanism is not discussed. A high concentration of irradiations-induced self-interstitial atoms could cause dislocation structures and further evolve into grain boundaries, thus lead to grain refinement (Acta Mater-2020-186-162). These possible mechanisms should be properly discussed.
6. In Fig. 8, how to distinguish whether it is a real grain fragmentation or actually an evolution of irradiation-induced dislocation loops?
7. In Fig. 1(g)-(h), the extra diffraction spots demonstrate there is a new phase in the irradiated samples. The new precipitated phase may be Cr-Hf rich particles since the irradiation temperature is close to the transition temperature in W_{29.4}Ta₄₂Cr₅V_{16.1}Hf_{7.5} alloy (see Fig. 6 and Fig. 7). However, in the ex-situ experiment (W₃₁Ta₃₄Cr₅V₂₇Hf_{3.0}), the Cr-V rich particles may exist in the irradiated sample due to negative SOR value around the irradiated temperature (see Fig. 6 and Fig. 7). Hence, the content of Hf has a great impact on the microstructure evolution after irradiation (as shown in Fig. 6). Are the properties of W₃₁Ta₃₄Cr₅V₂₇Hf_{3.0} and W_{29.4}Ta₄₂Cr₅V_{16.1}Hf_{7.5} alloys comparable?
8. The W-Ta-Cr-V-Hf alloy shows better irradiation resistance than previously studied W-Ta-Cr-V RHEA. The author should discuss the critical role of Hf element in-depth in this study.
9. The VEC value of current alloys is actually higher than 4.4, which does not meet the critical VEC for finding a better ductile alloy. This should be discussed.

The authors would like to thank the referees for their valuable comments and deep understanding of the subject. All our responses are in red below and changes to the manuscript highlighted yellow in the revised manuscript.

Reviewer #1 (Remarks to the Author):

Comment: This is work on a refractory HEA that has been designed through a combination of thermodynamic calculations together with experimental and modeling approaches for use in the harsh environments of fusion reactors. The material has been processed into thin films of two different thicknesses, and irradiation resistance, thermal stability, and strength were evaluated. The material appears to have excellent irradiation damage tolerance, and the obtained properties from one of the two thin films are correlating nicely with the simulations. The conducted work is of high quality and the study overall done nicely with the manuscript being written concisely and clearly. While as such the manuscript is overall suitable for publication in Nature Communications there are several drawbacks that need to be taken into account when a decision regarding acceptance is made. It can be assumed that Cr was reduced due to the formation of precipitates – if that's the case then this does not come out through.

We would like to thank the reviewer for his/her judgement. We appreciate the comments and we believe that now we have addressed them, as illustrated below, to improve the quality of the manuscript.

Reply: It is not decreasing the Cr content, but actually the addition of Hf which has a complex behavior as described in the paper. The addition of Hf tailor the SROs and the corresponding morphology. By adding Hf, the order-to-disorder transition temperature decreases and now we have not observed any precipitation at the same temperature where the old HEA does. More details are below when replying to the other comments.

Comment: One of the main questions is how novel this work really is? While the material appears to be a promising candidate for nuclear applications, the work is based on previous research on a WTaVCr alloy that has been published recently in Materials Today Energy and Science Advances and which has been modified through the addition of Hf based on established design principles for HEAs. Previous work on the 4-component alloy combined with the application of HEA design principles should not preclude publication;

Reply: We thank the reviewer for these comments. While the previous WTaCrV RHEA showed some radiation resistance, it suffered from Cr precipitation (induced by irradiation) and low ductility. The material studied here – i.e. the WTaCrVHf RHEA -- is totally novel and shows unmatched irradiation resistance (tested under dual beam and single beam conditions), morphology stability, lower ODTT and interesting phenomena (e.g. grain refinement under dual beam). We also have control over the morphology of the material (can alter ODTT). These combined with the methodology to increase ductility make this work highly novel. That said, it is worth emphasizing the past works we did on the WTaCrV RHEA are only precursors to this present study, which now introduces a completely new quinary WTaCrVHf RHEA with distinct response to both irradiation and high-temperature annealing as well as entirely different microstructure with even smaller grains at the nanoscale.

Comment: However, it is far from clear what's the purpose of processing two thin films of different thicknesses that are even in terms of composition quite different with particularly Ta, V, and Hf varying significantly (W29.4Ta42Cr5.0V16.1Hf7.5 vs. W31Ta34Cr5.0V27Hf3.0) and despite the author's comment about adding only minor additions of Hf, the final amount being comparable to Cr with 5%.

Reply: We thank the reviewer for this comment. The reason a thicker film was needed was to allow for nanoindentation, which was not possible on the thinner film. Additionally, we intended to have two different compositions to perform the simulations on the effect of Hf. As the reviewer knows it is very difficult to control the exact composition of a thin film via the use physical vapour deposition methods. Even if we follow the same deposition parameters during magnetron deposition, the thick film will have a (slightly) different composition. It is also important to point out that even with a change in the composition, both films had similar radiation resistance, which was one of the primary points of the paper: evaluate the effects of Hf in the response of the new quinary alloy to extreme environments. Additionally, having films with two different compositions did allow us to investigate if the ODTT can be modified with composition. Therefore, both thin films are herein studied under prototypic conditions, implying that the feasibility of synthesizing these alloys via other methods (*e.g.* arc melting) is outside the scope of this present study, but foster the community to engage in the protocol we developed to further study this W-Ta-Cr-V-Hf system.

Comment: While difficulties in preparation of two thin films with the same composition are certainly appreciated, making two thin films while then presenting results from mainly one of them is somewhat distracting – what's the point of this approach particularly if the irradiation campaigns are different? – are they even comparable?!

Reply: Thank you for these comments. All the HEAs with Hf demonstrated high irradiation resistance regardless of composition. To clarify this point, we have added the following to the paper:

“However, it should be noted that different compositions, tested in thin film forms, led to similar single-phase BCC microstructures with remarkably similar irradiation responses (in terms of dislocation loop and cavity damage as well as grain size stability), and for clarification, only one composition is discussed throughout this paper. Hence, the thick film composition follows similar irradiation resistance.”

Nevertheless, there were advantages for having different compositions as this allowed us to investigate if the ODTT can be modified with composition.

Comment: Irradiation depth certainly plays a role not just in terms of the irradiated region but also for the mechanical characterization – CSM measurements are good, but aren't they missing or at least just crossing the irradiated depth particularly when measurements from 400-500 nm depth are taken while the irradiated layer is 200-300 nm deep?

Reply: We apologize for this mistake. The measurements were actually taken from 200 nm depth and not 400-500 nm. There were two sentences on page 7 and one has been deleted to clarify this point.

Comment: These results, together with the statement that due to the large scatter in the nanoindentation data (which is known to be a problem with this method) ‘only reasonable results were shown’ are highly questionable!

Reply: There are many papers on nanoindentation. However, it is now standard to check the modulus along with the hardness to make sure that the nanoindentation is successful. For example, in pure W, we do many indentations, and we know the modulus of W and the shape of the modulus curve a-priori. If we get a modulus value far from the correct one (if known), or the curve shape is not correct, then we do not trust

this data. This is a methodology to select more reliable data from nanoindentation. We apologize for the confusion. We have now removed this statement since it is not a new thing we did but rather followed a standard protocol.

Comment: In terms of mechanical properties, it is also highly questionable what's the rationale for the authors recommending measuring ductility at larger grain sizes? Lower strength and larger ductilities at such larger grain sizes are almost guaranteed if the material could be processed using the same conditions and everything else being identical – this is, however, likely far from easy as the authors would/should have done this otherwise. If the material can be processed to larger dimensions, it has probably different composition and different phases.

Reply: This is a great comment by the reviewer. We really appreciate it. It is a hot topic that is being discussed currently in the HEA community. In general, it is well known thin films have nanometer grain sizes and that nanocrystalline materials lose ductility unless they are in the inverse Hall-Petch regime (below 20 nm for some material and 15 nm for other materials). Hence, we wanted to use the identified chemical composition from this protocol and manufacture bulk samples with the same composition. While the grain size in the bulk samples will be much larger than the nanocrystalline samples, we are hypothesizing that the chemical distribution, if processed appropriately, can stay homogeneous. However, this is a challenging problem.

Currently, additive manufacturing, hot pressing, sintering (with or without gas atomization or ball milling) deposition, and arc melting (with different mixing capabilities and techniques, and different currents) are all being studied. Every HEA should possess a recipe to be formed successfully. For example, Wei et al. (Nature Materials, 2020) has made a refractory high entropy alloy with as-cast ductility. Currently several projects are addressing these challenges.

As the reviewer stated, it is not easy, but possible with appropriate resources. This work here is about how the addition of Hf affects the morphology and irradiation resistance and about a rapid down selection protocol of refractory HEAs: thus it is a fundamental study using thin films that were processed and investigated under prototypic conditions. Our overall recommendation on larger grain sizes samples is to promote further research to enable feasibility for producing such RHEAs in their bulk forms, given their potential to be applied in extreme environments such as new fusion reactors. Therefore, following the principles of enhancing ductility was a criterion to follow as described in the supplemental.

In the paper we mention:

“The ductility enhancement can be examined (outside the scope of this paper) when the designed material is produced in homogeneous bulk and large grains form to examine the intrinsic ductility of the materials and avoid the loss in ductility in the nanocrystalline grain regime.[38]”

Comment: Next, why did the authors choose Hf instead of the other elements they mentioned as addition to their 4-component alloy and how did they measure SRO? In contrast to one of their statements, the impact of SRO on mechanical properties is highly questionable and it appears more and more that it is actually rather limited than affecting mechanical performance. In terms of the amount of volume change associated with irradiation and its impact on properties the authors contradict themselves at several locations in their work.

Reply: We chose Hf, as described in the paper and in more detail in the supplemental material, to minimize the VEC. Also, the element Hf presents low activation yield (*i.e.* it does not become highly radioactive when irradiated) for applications in neutron irradiation environments such as future fusion reactors. Furthermore, the addition of Hf was suggested in different works (cited in the manuscript) to enhance ductility.

The SRO is being considered here to check on the order to disorder transition temperature (ODTT) and intermetallic formation. It was not measured, but calculated. The SRO is related to the correlation functions depending on the system configuration. Such correlation functions are computed in the CE model at each temperature and from them the SRO is calculated. The referee is invited to check ref 51 in the manuscript (J. Phase Equilib. Diffus. (2017) 38:391–403) where the expressions are described.

In the manuscript, we write:

“The Warren-Cowley short-range order (SRO) parameters are used to quantify the chemical ordering between pairs of different species up to the second nearest neighbor by:

$$\alpha_n^{ij} = 1 - \frac{\psi_n^{ij}}{c_i c_j}$$

Where α_n^{ij} is the chemical short-range order parameter, ψ_n^{ij} is the probability of two atoms within n^{th} neighbor shell and c_i and c_j are the concentrations of species i and j , respectively. ψ_n^{ij} is calculated through a matrix inversion of correlation functions obtained via Canonical Monte Carlo (CMC). [51]”

The modification of the SRO affects ODTT. We want the material with the lowest ODTT. We have not discussed the effect of SRO on the mechanical properties. We discuss SRO as a parameter that affects morphology, and not as mechanism or preference.

In the paper we wrote:

“The chemical short-range order parameters (SRO) as a function of temperature as calculated from a 16000 atom simulation cells of the HEAs is plotted in Figures 6 and 7, respectively to discuss the SRO effect on the material morphology.

Comment: Despite these raised issues, it should be stated that the work is, overall, quite nice but the presented approach that is described as material design protocol likely overstated.”

Reply: We appreciate your comments. Regarding calling it a design protocol, it is because with the methodology involved here, we can design (to follow a certain criterion) rapid test and understand the morphology and irradiation resistance performance of any refractory HEA. The simulations performed, and the experimental validation can be applied to any system. An example is the Cluster Expansion (CE) simulations, which were optimized and validated with Atomic Probe Tomography (APT) and Transmission Electron Microscopy (TEM). Now the methodology involved in the CE can be applied to any other refractory BCC system.

Reviewer #2 (Remarks to the Author):

Comment: This work is essentially an extension of their previous work published in Science Advances 2019. The quality of the work is high, and the results are solid. The reviewer however has some major concerns before recommending its publication.

Reply: We would like to thank the reviewer for their comments. We appreciate the comments, and believe that we have addressed them, as illustrated below, to make our manuscript stronger.

Comment: Most importantly, the logic of the storyline is not understandable to the reviewer: so they follow the criteria to design ductile RHEAs, but they only intend to verify the ductility in a separate work? This does not make any sense, does it? How can we know for sure that the design strategy works or not? As a matter of fact, the super-high hardness almost for sure indicates the material is brittle. They also used the wording of ductility enhancement for the pending work, making a hint that the ductility might be there in bulk samples with large grain size. Of course, it cannot be said for sure that it would not happen. However, lacking any evidence of ductility, the story is certainly not self-consistent.

Reply: We thank the reviewer for their comment. There are two goals in this paper. The first goal is to establish a protocol to design new HEAs and rapidly down-select the best compositions: A design protocol that consists of thermophysical parameter calculations, Calphad simulations and Cluster Expansion/Monte Carlo simulations in addition to the rapid testing with in-situ and ex situ techniques. However, this design considered minimizing the VEC concentration and single-phase criterion for ductility purposes. The second goal is to show the superior performance of the new HEA in terms of irradiation resistance to single and dual beam, the corresponding phenomena (e.g. grain refinement) and the ability to tune the ODTT by changing the composition in this material system.

Testing the ductility should/will occur after the successful manufacturing process of the bulk form of these HEAs.

To test ductility, we need 1) to manufacture a bulk material with elements that are homogenously distributed (similar morphology in terms of distribution to the deposited samples), and 2) avoid being in nanocrystalline regime. It is known that nanocrystalline materials lose ductility unless they are in the inverse Hall-Petch regime (below 20 nm for some material and 15 nm for other materials). If there is segregation of the elements in the bulk samples, the material will fail in a brittle manner. However, this brittleness is not intrinsic. It is due to the manufacturing methodology being not optimized or successful. In the paper we wrote:

“In this study, we have developed a design strategy to further improve the overall response of the W-Ta-Cr-V RHEA. The aim of the design is to develop a material which higher irradiation resistance, high thermal stability, enhanced control over the morphology and no precipitation at reactor-relevant temperatures while following the criteria for enhanced ductility suggested in literature.[30] It has been shown that BCC RHEAs can show enhanced intrinsic ductility with elongation between 6 and 15% if the valence electron concentration (VEC) remains below 4.4... ..”

Currently, additive manufacturing, hot pressing, sintering (with gas atomization or ball milling) deposition, and arc melting (with different mixing capabilities and currents) are all being studied. Every HEA should

have a recipe to be grown with specific theoretical mechanical properties. For example, Wei et al. (Nature Materials, 2020) has made a refractory high entropy alloy with as-cast ductility. His recipe and results warranted a Nature Materials publication. Without this recipe, the material is brittle. Currently several projects are funded nationally to achieve this purpose. To test this material system fairly, we need to achieve homogeneous bulk forms of these HEAs which is a different project. We know now that this material has unmatched irradiation resistance (e.g. very low change in volume for 9% He implantation). Now the focus can be in ductility which will require a successful manufacturing and other metallurgy principles such as grain orientation, stress minimization and homogenous distribution (similar to the deposited films) at large scales.

Comment: They have such a statement: the equilibrium structure of Hf is hexagonal-closed packed (HCP) versus BCC for the other elemental components of the alloy. Hence, only small quantities of Hf can be added in order to retain a single-phase BCC structure. This statement is not true. Indeed, the Senkov alloy, equiatomic HfNbTaTiZr, has a single-phase BCC structure and contains high quantities of Hf, Ti and Zr, all of HCP structure. Therefore, the statement cannot be justified.

Reply: We appreciate this comment. That sentence has been removed. The amount of Hf that can result in phase change depends on the material system. However, we agree with the reviewer opinion on the statement was not worded appropriately and it has been removed.

Comment: They gave two compositions in thin and thick films, and they also admitted that both produced films have different compositions. However, they never gave the designed compositions or compositional range. What is indeed the product (compositions) of the design strategy, i.e., to design ductile RHEAs?

Reply: We tested several thin films of different compositions within the quinary W-Ta-Cr-V-Hf system (as mentioned in the manuscript). All have similar radiation resistance. The difference in composition is a plus as we aimed at studying whether different compositions in the system could lead to different morphologies as a function of temperature (e.g. SROs, precipitations, etc), and if compositional variations can lower the ODTT value, which is a very important output for the design of new RHEAs. The supplemental material has many calculations on different compositions trying to minimize the VEC as much as possible, thus addressing on the reviewer's comment on the compositional range. Different compositions can affect dislocation generation and migration energies and therefore, ductility must be studied for different compositions that have low ODTT, high thermal stability and excellent radiation resistance (output of this paper). That said, we conclude that the product of the design strategy were the two alloys we propose (both thin and thicker film) as we directly assessed its superior radiation tolerance.

Comment: There is such a statement: the He average migration energy was found to be 0.156 eV in the alloy compared to 0.06 to 0.081 eV in pure W and the formation energy was ~2 times lower. Where does this 2 times lower formation energy come from?

Reply: This is for the 4-material system. We computed the formation energy of He in different RHEA configurations using density functional theory. The average formation energy that we obtained was 3.57 eV which is about half of the formation energy in pure W, in the range 6.160 - 6.365 eV. It is published in El Atwani et al. Materials Today Energy, 19, 100599, 2021.

Comment: The hardness increases after annealing and irradiation are way too significant: 3GPa and 3.75GPa, respectively. They used compositional segregation to account for the former, and segregation,

grain refinement plus lattice distortion to account for the latter. This qualitative is not convincing enough, and a quantitative analysis is necessary here.

Reply: We agree with the reviewer. All the factors listed above are playing a role. Deconvolution of these phenomena is very complex topic of research. Grain refinement, grain boundary segregation, invisible clusters due to irradiation, and lattice distortion all do affect hardness. This is an HEA and requires isolating these effects to come up with an answer. If it was a pure material with no grain refinement, then it should have irradiation defects. For example, Zheng et al. Atomic-Scale Hidden Point-Defect Complexes Induce Ultrahigh-Irradiation Hardening in TungstenNano. Lett. 2021) has performed simulations together with experimental work to understand which defects contribute to hardening in pure W (helium bubbles that are visible in the microscope or the invisible He-V complexes). In an HEA, additional phenomena occur, and science is still progressing in these aspects. We added one sentence in the paper to mention that deconvolution of these effects is complex in a HEA system. In the paper we wrote:

“The 3.75 GPA difference at 200 nm depth should include the elemental segregation (although it decreases after irradiation as shown in Figure 2), grain refinement due to irradiation, invisible clusters in the microscope, and the enhancement in lattice distortion due to homogenization of the elements. However, deconvolution of these effects is a complex process in a HEA system where several competing phenomena occur. Nevertheless, it is expected that this work will promote further studies on understanding this complex behavior of HEAs exposed to extreme irradiation conditions.”

Reviewer #3:

Comment: This study developed a new strategy to further improve the performance of W-based high entropy alloy via microstructural design according to the combination of valence electron concentration analysis, phase diagram calculation and density-functional theory simulation. According to the predictions, some of the compositions were experimentally prepared and tested. Based on the new protocol, the manuscript designed a promising W-Ta-Cr-V-Hf alloy with excellent properties, including irradiation resistance, thermal stability, and strength. The agreement between experimental and modeling results provides a valid strategy for designing high performance materials for extreme environmental applications. This is a nice manuscript with sufficient novelty for publication in Nature Communications.

Reply: We would like to thank the reviewer for his/her appreciative assessment of our submission. We appreciate the comments and believe that we have addressed them all. As illustrated below, your comments significantly improved our manuscript.

Some technical comments are listed below:

Comment: The protocol predicts a better ductility of W-Ta-Cr-V-Hf alloy. Is it possible to prepare bulk nanocrystalline alloy with the same composition? Do you expect a similar performance in the bulk nanocrystalline W-Ta-Cr-V-Hf alloy? Is it brittle?

Reply: This is a great question. We manufactured the material using magnetron sputtering which provides uniform composition. In general, it is well known thin films have nanometer grain sizes and that nanocrystalline materials lose ductility unless they are in the inverse Hall-Petch regime (below 20 nm for some material and 15 nm for other materials). Hence, we want to use the identified chemical composition

from this protocol and manufacture bulk samples with the same composition. While the grain size in the bulk samples will be much larger than the nanocrystalline samples, we are hypothesizing that the chemical distribution, if processed appropriately, can stay homogeneous. However, this is historically a challenging manufacturing problem within the field of refractory alloys. If the bulk samples have elemental segregation, it will affect ductility regardless of grain size.

Comment: Is there any limitation in current protocol? Whether is it valid for all other HEA alloys design? It is better to discuss these points in the revised manuscript.

Reply: The protocol consists of thermophysical parameter calculations, Calphad, CE and Monte Carlo simulations to predict the morphology and ODTT, and rapid characterization of the materials under in situ tools. There are no limitations on the type of HEA or the elements. To reflect on this, in the manuscript we mention:

“The established design protocol can be further utilized to design and synthesize new RHEAs and constitutes a material design paradigm with high throughput morphology predictions.”

Comment: Is there any surface effect in the in-situ irradiation experiment of W-Ta-Cr-V-Hf alloy? For normal metals, a strong surface sink effect would influence the species and concentration of defects. It is better to explain.

Reply: It is documented before that surface effects vanish when grain boundary to surface ratio approaches the value of 1. (El Atwani et al. Unprecedented irradiation resistance of nanocrystalline tungsten with equiaxed nanocrystalline grains to dislocation loop accumulation, Acta 2019, Li et al. Study of defect evolution by TEM with in situ ion irradiation and coordinated modeling, Phil. Mag, 2012)

The HEAs in this work have a ratio that is over 1. The grain sizes are approximately 25 nm which means we have four grains in through the thickness of the sample. Moreover, the sluggish diffusion in HEAs further limits any surface effect. In the manuscript, we have added the following to clarify this point:

“It should be noted that possible surface effects are expected to diminish when the grain boundary to surface ratio approaches the value of 1.[67,68] In this HEA, the surface ratio is approximately 10. Furthermore, surface effects in the HEA system is expected to be smaller than in pure W due to the rougher defect migration landscapes [69], which can be applicable to some HEAs. [70]”

Comment: In Fig. 5, can author explain the reason why the hardness show a turning point in the depth of 0~200 nm for the irradiated and the annealed alloys? Why is the irradiated sample harder than the annealed sample in the deeper region (400~800 nm)?

Reply: We apologize that this was not clear. However, we do provide some reasons behind this in the manuscript:

“In Figure 4, the irradiated curve approached the annealed curve at high depths that exceed the irradiation depth by a factor of ~ 3 suggesting that the change in hardness (~ 1.6 GPa) at that depth is due to elemental segregation at the grain boundaries after annealing. The 3.75 GPa difference at 200 nm depth should include the elemental segregation (although it decreases after irradiation as shown in Figure 2), grain refinement due to irradiation and the enhancement in lattice distortion due to homogenization of the elements.”

Comment: What is the mechanism of irradiation hardening? The authors mentioned that materials homogenization and grain refinement are two main reasons. However, a recent study shows that numerous invisible He-V clusters are the main hardening contributor in irradiated pure W (Nano Lett-2021-5798).

Reply: Great comment and question. We now mention in the manuscript:

“Small defect clusters which are not visible in the TEM can affect the hardness. However, considering the dispersed barrier hardening (DBH) model [79] and other experimental work in W [80], the increase in hardness in this HEA cannot all be justified by the invisible defect clusters in TEM.”

Certainly, these complexes can affect, but cannot justify the large change in hardness. In the paper suggested by the reviewer, it was found that He_n-V complexes can result in low peak stresses when n is 1. In our material, we expect to have several vacancies per helium (low pressure) and therefore, we are in the regime where He-V complexes affect the results but cannot justify the large change observed. Without knowing alpha (the barrier strength) in our material, we cannot predict a more accurate contribution for the He-V complexes. The paper is cited as it strengthens our argument that the invisible complexes can affect the results.

Comment: In addition, what is the mechanism of grain refinement? It is proposed that irradiation-induced grain fragmentation cause the grain refinement but the underlying mechanism is not discussed. A high concentration of irradiations-induced self-interstitial atoms could cause dislocation structures and further evolve into grain boundaries, thus lead to grain refinement (Acta Mater-2020-186-162). These possible mechanisms should be properly discussed.

Reply: Grain refinement was discussed in the paper on pages 20 and 21 in detail. The suggested mechanism falls into the first category discussed on page 20

Comment: In Fig. 8, how to distinguish whether it is a real grain fragmentation or actually an evolution of irradiation-induced dislocation loops?

Reply: The video shows fragmentation and no loop formation. We see a grain that was fragmented. In addition, fast tilting done shows that these changes we see are not loops. Loops which are sensitive to g.b will appear and disappear depending on the g.b condition.

Comment: In Fig. 1(g)-(h), the extra diffraction spots demonstrate there is a new phase in the irradiated samples. The new precipitated phase may be Cr-Hf rich particles since the irradiation temperature is close to the transition temperature in W_{29.4}Ta₄₂Cr₅V_{16.1}Hf_{7.5} alloy (see Fig. 6 and Fig. 7). However, in the ex-situ experiment (W₃₁Ta₃₄Cr₅V₂₇Hf_{3.0}), the Cr-V rich particles may exist in the irradiated sample due to negative SOR value around the irradiated temperature (see Fig. 6 and Fig. 7). Hence, the content of Hf has a great impact on the microstructure evolution after irradiation (as shown in Fig. 6). Are the properties of W₃₁Ta₃₄Cr₅V₂₇Hf_{3.0} and W_{29.4}Ta₄₂Cr₅V_{16.1}Hf_{7.5} alloys comparable?

Reply: We appreciate the deep thinking and comment by the reviewer. In terms of morphology, they differ in the order to disorder transition temperature. In terms of radiation resistance, as mentioned in the paper, several compositions were studied, and no loop formation occurred. In terms of mechanical properties only the thick film was studied.

When we discussed the extra spots, we considered the modeling (Figure 7) and the APT validation (Figure 9). APT at that temperature, showed homogenous compositions and the same should happen after irradiation. However, we model the matrices and not the grain boundaries. The extra rings should belong to the Hf on the grain boundaries. What we thought, is that if Cr-Hf are present in the in-situ sample, they should also show up in the ex-situ sample which has a larger ODTT. Also, the Cr-Hf was not seen in the TEM images or the EDX maps except Hf and Cr on the grain boundaries.

Comment: The W-Ta-Cr-V-Hf alloy shows better irradiation resistance than previously studied W-Ta-Cr-V RHEA. The author should discuss the critical role of Hf element in-depth in this study.

Reply: The Hf addition is being studied in Figure 6 and Figure 7. New binaries (enthalpy of mixing) were calculated, and Monte Carlo simulations were performed to understand Hf effects on the thermodynamic properties of the system. Two different concentrations of Hf were used to demonstrate the effect further. The supplemental material has calculations and illustrations to demonstrate the effect of Hf.

Comment: The VEC value of current alloys is actually higher than 4.4, which does not meet the critical VEC for finding a better ductile alloy. This should be discussed.

Reply: That is true. The first few pages in the Results section illustrate that. We followed the criteria of minimizing VEC and have a single BCC phase to come up with a material and establish the design protocol. The VEC criteria is based on experimental results (small database) in literature, and more experimental results and works are needed to make sure that the number mentioned in the literature now fits all HEAs. However, minimizing VEC is one possible strategy and we followed it here.

Reviewers' comments:

Reviewer #1 (Remarks to the Author):

While it is not appreciated that the authors dissect the comments of the original review thereby making it unnecessarily complicated to follow the logic of the original response, they have overall given reasonable comments to the various points of the previous review. Stating that the addition of few % Hf to a previously studied material composition makes this an entirely new and novel study is nonsense so that the judgement about the lack of novelty remains an issue. Furthermore, the issue of large-scale processing of such material remains unchanged – this is, however, something everybody working on these materials faces and as such a drawback that is outside the scope and control of this work.

Reviewer #2 (Remarks to the Author):

The reviewer is not satisfied with the provided response regarding ductility. They intend to produce ductile RHEAs, but they are now certainly producing brittle thin films. This could be due to size effect, which is understandable. How to verify the ductility then? Using bulk samples. Right? Bulk samples are not prepared, and cannot be tested. No, this is not a convincing story at all. They were just trying to explain why ductility could not even there in bulk samples because of reasons like segregation (grain size is certainly not an issue here). Let's put it this way: if the "ideal" bulk sample is indeed that difficult to be produced, what's the point then?

The reviewer is also not satisfied with the interpretation to the abnormal hardness increase. In the revised version, they simply added a few more possible mechanisms. How can that be helpful? A complex issue? Of course! Isn't providing the solutions to complex issues exactly what is expected from prestigious journals like Nature Communications? And, regarding the newly added mechanisms for the abnormal hardness increase: the enhancement in lattice distortion due to homogenization of the elements? How come? That certainly deserves a much better justification, than just saying it.

Reviewer #3 (Remarks to the Author):

The revisions overall look satisfactory. One last comment about the ductility of the alloy. The authors use a large paragraph to state the design of the alloy is following the criteria for enhanced ductility based on the valence electron concentration below 4.4. However, the new designed alloy actually does not meet this criteria since the VEC is much larger than 4.4. Furthermore, the ductility of the alloy is not tested at all in this study, and even it is in homogeneous bulk and large grains form, can you guarantee it will have improved ductility? In the revised manuscript, the authors state that the ductility enhancement is outside the scope of this paper, but the design of the alloy is following the ductility criteria, such a contradiction sounds a little bit strange. The authors are better to adjust this logic and lower the voice on the ductility of the alloy.

Reviewer #1 (Remarks to the Author):

Reviewer: While it is not appreciated that the authors dissect the comments of the original review thereby making it unnecessarily complicated to follow the logic of the original response, they have overall given reasonable comments to the various points of the previous review. Stating that the addition of few % Hf to a previously studied material composition makes this an entirely new and novel study is nonsense so that the judgement about the lack of novelty remains an issue. Furthermore, the issue of large-scale processing of such material remains unchanged – this is, however, something everybody working on these materials faces and as such a drawback that is outside the scope and control of this work.

Response: We tried in the response to be detailed as much as we can to clarify every point the reviewer raised. Dissecting the comments, in the previous review, was to make things clearer. Probably, it made things complicated which was opposite to our goal. We would like to note that the addition of Hf made it another new alloy with different properties, morphology behavior as a function of temperature, irradiation resistance, etc. There is research performed on W-Ta-V. However, our W-Ta-V-Cr demonstrated different outcomes. The addition of 0.5% TiC to W demonstrated very different properties and performance. Few percent changes of some elements in steels results in different performances and names (e.g., HT-9 steel vs Grade 91 vs Grade 92, etc.). We understand the reviewer's concern regarding novelty, however, and we agree that we need to make changes and clarify that. We believe that the changes are important to improve the quality of our paper and reflect on the reviewer's concern.

The paper is not about bulk manufacturing, and as the reviewer noted, many are working on this issue. We use films as a rapid downselection process. Bulk manufacturing of these alloys of the same quality, chemistry, and elemental distribution to films (which is the ideal case) is possible (and some already succeeded to find the recipe for their alloys, e.g., *Nature Materials*, 19, 1175-1181, 2020) but requires extensive works and funds which is not the scope of this paper. The scope here is to design an alloy system, rapid downselect compositions and investigation the outstanding performance of these compositions.

We now made changes by re-writing sections, removing sections, changing a figure, and changing the title.

Reviewer #2 (Remarks to the Author):

Reviewer: The reviewer is not satisfied with the provided response regarding ductility. They intend to produce ductile RHEAs, but they are now certainly producing brittle thin films. This could be due to size effect, which is understandable. How to verify the ductility then? Using bulk samples. Right? Bulk samples are not prepared, and cannot be tested. No, this is not a convincing story at all. They were just trying to explain why ductility could not even there in bulk samples because of reasons like

segregation (grain size is certainly not an issue here). Let's put it this way: if the "ideal" bulk sample is indeed that difficult to be produced, what's the point then?

The reviewer is also not satisfied with the interpretation to the abnormal hardness increase. In the revised version, they simply added a few more possible mechanisms. How can that be helpful? A complex issue? Of course! Isn't providing the solutions to complex issues exactly what is expected from prestigious journals like Nature Communications? And, regarding the newly added mechanisms for the abnormal hardness increase: the enhancement in lattice distortion due to homogenization of the elements? How come? That certainly deserves a much better justification, than just saying it.

Response: We do think that our writing style in the previous version has led to misunderstandings. We agree that the reviewer's comments are important. However, ductility is out of scope. We use films to rapidly design, downselect, optimize and fundamentally understand materials. Bulk manufacturing is another project that needs time, effort, funds, etc.

We had three screening methodologies to design the alloy: 1- Thermophysical parameter calculations to ensure solid solution and to minimize the VEC. To minimize the VEC, several elements and strategies are possible but the strategy of adding Hf is chosen for several purposes mentioned in the manuscript. 2- Calphad with the newest database. 3- Cluster Expansion and Monte Carlo simulations to predict the final morphology as a function of temperature. After that, the performance regarding irradiation resistance, the phenomena observed, thermal stability in terms of segregation and grain size change, hardness, and the comparison with modelling to demonstrate how we can predict morphologies, are demonstrated. We only mentioned that we want to minimize the VEC because it is suggested to improve dislocation dynamics and ductility in some HEAs.

Manufacturing of these materials in bulk with similar chemistry and elemental distribution to films is possible, but this is a different project. Manufacturing, optimization, and assessment of a bulk HEA is a work that probably should result, by itself, in a paper much longer than what Nat. Comm accepts. Making HEAs in bulk forms with ideal microstructure (predicted by theory and demonstrated in thin and thick films) is a project and not a material that can be purchased or a material with known recipe. For example, our group has received an ARPA E 2022 award of 3.2M for 3 years to make this material in bulk using additive manufacturing. Several institutions are involved. The point of this work is to invent a material with known composition and chemistry and with great performance (probably Technology Readiness Level, TRL of 2-3). Most of graphene related works, for example, belong to this TRL and many projects are trying to make devices from these materials. For our material, jumping from TRL 3 to 5 is another project.

To address the point of the reviewer, we deleted sections and re-wrote sections, and the paper now has no relation with ductility. Maybe our writing style in the previous version, has raised confusions. We focus on the data we have, and the scope intended for this work.

Regarding the hardness, there are several projects now running in different groups to understand the parameters affecting hardness increase in HEAs vs pure metals. Over 6 factors are involved and there are synergistic effects among these factors.

To address the point of the reviewer, we have analyzed the data further and we modified the figure and the discussion to mention the differences between all curves. Synergistic effects among factors remain, however. The effect of TEM-visible defects and TEM-invisible defects are yet to be fully understood and resolved in pure metals after many years of research. We now removed any speculation, and references are added when referring to outcomes.

Reviewer #3 (Remarks to the Author):

Reviewer: The revisions overall look satisfactory. One last comment about the ductility of the alloy. The authors use a large paragraph to state the design of the alloy is following the criteria for enhanced ductility based on the valence electron concentration below 4.4. However, the new designed alloy actually does not meet this criteria since the VEC is much larger than 4.4. Furthermore, the ductility of the alloy is not tested at all in this study, and even it is in homogeneous bulk and large grains form, can you guarantee it will have improved ductility? In the revised manuscript, the authors state that the ductility enhancement is outside the scope of this paper, but the design of the alloy is following the ductility criteria, such a contradiction sounds a little bit strange. The authors are better to adjust this logic and lower the voice on the ductility of the alloy.

Response: The reviewer had one more comment about ductility which is an important aspect of a material's final design. However, our paper was never about ductility. Maybe, our writing style has raised confusions and we agree with the reviewer. We had three screening methodologies to design the alloy: 1- Thermophysical parameter calculations to ensure solid solution and to minimize the VEC. To minimize the VEC, several elements and strategies are possible but the strategy of adding Hf is chosen for several purposes mentioned in the manuscript. 2- Calphad with the newest database. 3- Cluster Expansion and Monte Carlo simulations to predict the final morphology as a function of temperature. After that, the performance regarding irradiation resistance, the phenomena observed, thermal stability in terms of segregation, hardness, and the comparison with modelling to demonstrate how we can predict morphologies, are demonstrated. We only mentioned that we want to minimize the VEC because it is suggested to improve dislocation dynamics and ductility. The 4.4 value of the VEC is taken from few works on HEAs. The 4.4 value may not be applicable to all alloys. The morphology, phases, and other parameters can affect this value. It is not an absolute value. Therefore, we only aimed to minimize the VEC.

To reflect on the reviewer's point, we changed the title, removed sections, and re-wrote sections to clarify our design steps. The paper now has no relation with ductility. We focus on the data we have, and the scope intended for this work.

REVIEWERS' COMMENTS

Reviewer #2 (Remarks to the Author):

Unfortunately, the authors are simply trying to avoid the questions, and not trying to address to them. The logic is that, with the current level of work, it is already good enough to be published at Nature Communications. The other parts, if done in the future, can lead to more Nature Communications works or some other top journal entries.

Every gradual improvement can then be marketed as a rather novel discovery, as seen in this case. To the reviewer, that is not acceptable.

Reviewer #3 (Remarks to the Author):

The authors have removed the controversial discussions related to ductility. The revised manuscript reads smooth and is suitable for publication.